# Adolescent and adult mice use both incremental reinforcement learning and short term memory when learning concurrent stimulus-action associations

**Juliana Chase** [1,2☯], **Liyu Xia** [3☯], **Lung-Hao Tai** [2], **Wan Chen Lin** [1,2], **Anne G. E. Collins** [1,4‡]*, **Linda Wilbrecht** [1,2‡]*

**1** Department of Psychology, University of California, Berkeley, Berkeley, California, United States of America, **2** Department of Neuroscience, University of California, Berkeley, Berkeley, California, United States of America, **3** Department of Mathematics, University of California, Berkeley, Berkeley, California, United States of America, **4** Helen Wills Neuroscience Institute, University of California, Berkeley, Berkeley, California, United States of America

☯ These authors contributed equally to this work.
‡These authors also contributed equally to this work.
* annecollins@berkeley.edu (AGEC); wilbrecht@berkeley.edu (LW)

**Data Availability Statement:** All data (raw, processed, and simulated) is publicly available with an associated DOI at https://osf.io/ue83f/ and

## Abstract

Computational modeling has revealed that human research participants use both rapid working memory (WM) and incremental reinforcement learning (RL) (RL+WM) to solve a simple instrumental learning task, relying on WM when the number of stimuli is small and supplementing with RL when the number of stimuli exceeds WM capacity. Inspired by this work, we examined which learning systems and strategies are used by adolescent and adult mice when they first acquire a conditional associative learning task. In a version of the human RL+WM task translated for rodents, mice were required to associate odor stimuli (from a set of 2 or 4 odors) with a left or right port to receive reward. Using logistic regression and computational models to analyze the first 200 trials per odor, we determined that mice used both incremental RL and stimulus-insensitive, one-back strategies to solve the task. While these one-back strategies may be a simple form of short-term or working memory, they did not approximate the boost to learning performance that has been observed in human participants using WM in a comparable task. Adolescent and adult mice also showed comparable performance, with no change in learning rate or softmax beta parameters with adolescent development and task experience. However, reliance on a one-back perseverative, win-stay strategy increased with development in males in both odor set sizes, but was not dependent on gonadal hormones. Our findings advance a simple conditional associative learning task and new models to enable the isolation and quantification of reinforcement learning alongside other strategies mice use while learning to associate stimuli with rewards within a single behavioral session. These data and methods can inform and aid comparative study of reinforcement learning across species.

supporting source code, analyses, and modeling and helper functions is available online at https://github.com/Wilbrecht-Lab/mouse_RLWM.

**Funding:** This work was funded by NSF SL-CN: Science of Learning in Adolescence: Integrating Developmental Studies in Animals and Humans (Award #1640885 to L.W. and A.G.E.C.) and the Simons SFARI Foundation (award #613972 to L. W.). The funders had no role in study design, data collection and analysis, decision to publish, or preparation of the manuscript. www.nsf.gov; www.sfari.org.

**Competing interests:** The authors have declared that no competing interests exist.

## Author summary

Here we studied the strategies and mechanisms mice use to learn a simple two choice odor based task in a single session in late stages of development. Using a set size manipulation and computational models we find evidence that mice use incremental reinforcement learning as well as several short-term (one-back) strategies to earn water reward. Our data and models clarify how mice learn a simple task and establish methods by which mouse and human reinforcement learning may be isolated for cross-species comparison of learning.

## Introduction

Mice are the species of choice in many neuroscience experiments due to their genetic tractability, yet questions remain regarding their value as a model for human brain function. As cognitive dysfunction and differences in learning are increasingly implicated in a range of psychiatric and neurodevelopmental disorders, it is essential to understand if mice rely on similar processes as humans to calibrate mouse to human translation. Here, we studied if mice use short-term memory or other strategies alongside reinforcement learning when learning new stimuli-action associations in an odor-based two alternative forced choice (2AFC) task.

There is positive evidence that rodents have working memory, an effortful and flexible form of short-term memory typically associated with higher level executive functions in humans [1]. Rodents can solve tasks that require correct responses to a cue after a delay lasting dozens of seconds [2]. For example, in studies of spontaneous alternation and delay-non-match-to-sample, rodents are able to remember where they were seconds to minutes before and change their response accordingly [3]. In these same tasks, there is evidence for increases in rodent working memory in the transition from adolescent to young adult [4–7]. Despite this evidence, it is unknown if rodents rely on working memory during simple reinforcement learning tasks where they are rewarded for learning cue-action associations.

It has been noted that in human participants working memory can play a leading role in learning, even in simple tasks which are not intended to tap working memory. Recent evidence has highlighted that much of what is typically considered "reinforcement learning" behavior, where participants learn cue-action associations from rewards, actually relies on working memory processes [8, 9]. In this work, rapid, yet capacity limited working memory (WM) processes were shown to be used first to solve the task (i.e. when learning about two cues in a block) while incremental but robust reinforcement learning (RL) processes were only additionally revealed when working memory's limited capacity was exceeded (i.e. when learning more than 3–5 cues in a block, depending on individuals) [8]. The discovery of contributions of WM to RL was facilitated by the design of a trial-by-trial learning task with set size (the number of stimuli in a block) manipulation and a novel computational model called RL+WM. The task design and model can be used to estimate learning supported by either WM related processes or RL processes in different set sizes [8]. In a developmental study of human subjects aged 8–30 using the RL+WM task and model, both WM and RL metrics were found to change during adolescent development [10], changes likely reflecting the maturation of both the prefrontal cortex and the basal ganglia [11].

Here, we adapted the RL+WM task for mice in order to test if 1) working memory is used alongside reinforcement learning in mice during cue-action learning, and whether 2) mice change how they learn the task during adolescent development.

Based on the above data from the human and rodent literature, we predicted that we would see a signature of WM use in mice while learning at low, but not high, set sizes in a rodent version of the RL+WM task. We also hypothesized that the reliance on WM in learning would increase throughout adolescent development, based on past rodent and human work [4–7, 10].

We trained mice in an odor based RL+WM task, in which mice successfully learned new odors in multiple set sizes and within a single session at all ages tested. Following behavioral and regression analyses, we developed a series of models outside of RL+WM to examine unique features of mouse behavior. Based on these models, we concluded that mice use basic incremental RL processes along with simple "one-back" strategies at the policy level to solve the task, potentially suggesting a primitive form of short-term memory dependence in learning that falls short of human-like working memory. Some one-back strategies became more commonly used with age in males while the use of others changed significantly over sessions, suggesting a role of sex, development, and experience in shaping short-term strategies. Overall, these data reveal important species differences in neuro-cognitive systems used for learning a simple task and suggest methods for isolating more comparable processes for cross-species comparison.

## Results

### Behavioral task and results

Mice were trained in the RL+WM task using novel odors at ages ranging from P30 to P90. Following a brief habituation period, mice entered the early training phase where they were trained with the first set of odors "A & B," delivered at the central initiation port with a left choice rewarded for odor A and right choice rewarded for odor B (Fig 1A). After two to three sessions in the early training phase, mice then experienced novel odors presented in set sizes of either n = 2 or n = 4 in the RL+WM task (novel odor phase) (Fig 1A and 1B). Each mouse experienced at least 3 sessions of each set size, interleaved, with set size = 2 always experienced first. Performance "readiness" was checked at the beginning of each session by requiring mice to reach 70% performance for both A & B within 200 trials prior to moving on to novel odors (Fig 1B).

In the novel odor learning phase of RL+WM, we found evidence of significant learning within a single session in both males and females of all ages. For each new odor, mice started near chance performance, but over the course of two hundred trials per odor (binned by 25 presentations of each) they approached an average of 70% performance (Fig 2A and 2D).

In humans performing RL+WM, stimuli encountered in lower set sizes (e.g. 2 cues) have been shown to be learned quickly using WM, while stimuli in higher set sizes (e.g. 6 cues) are learned more gradually with greater reliance on RL systems. In both male and female mice (all ages combined), we found that learning was similarly gradual for odors presented in set sizes of two or four odors (Fig 2A and 2D). Male mice had a small but significant difference in performance between set size = 2 and set size = 4 as well as in the interaction between set size and time that emerged gradually over the course of the session (Fig 2A, RM 2-way ANOVA, set size: $F(1, 205) = 4.12$, $p = 0.04$, set size $^*$ time interaction: $F(7, 1435) = 2.43$, $p = 0.01$). The weak set size effect in males favoring set size = 2 could be due to the fact that stimuli are repeated more often in set size 2. Notably, in non-repeated trials, fraction correct was higher in set size 4 compared to set size 2. (Fig 2B: Mann-Whitney $U = 7507$, $p < 0.0001$). This could be because in set size 4, but not set size 2, the same action is correct for half the stimuli, even when the stimulus has changed. Thus, although we saw set size differences in males, this may be due to the success of repeating a choice rather than a signature of WM.

Males and females had similar behavioral performance in the task overall (Mann-Whitney, $U = 14046$, $p = 0.19$), both showing stronger performance in repeat versus non-repeated trials

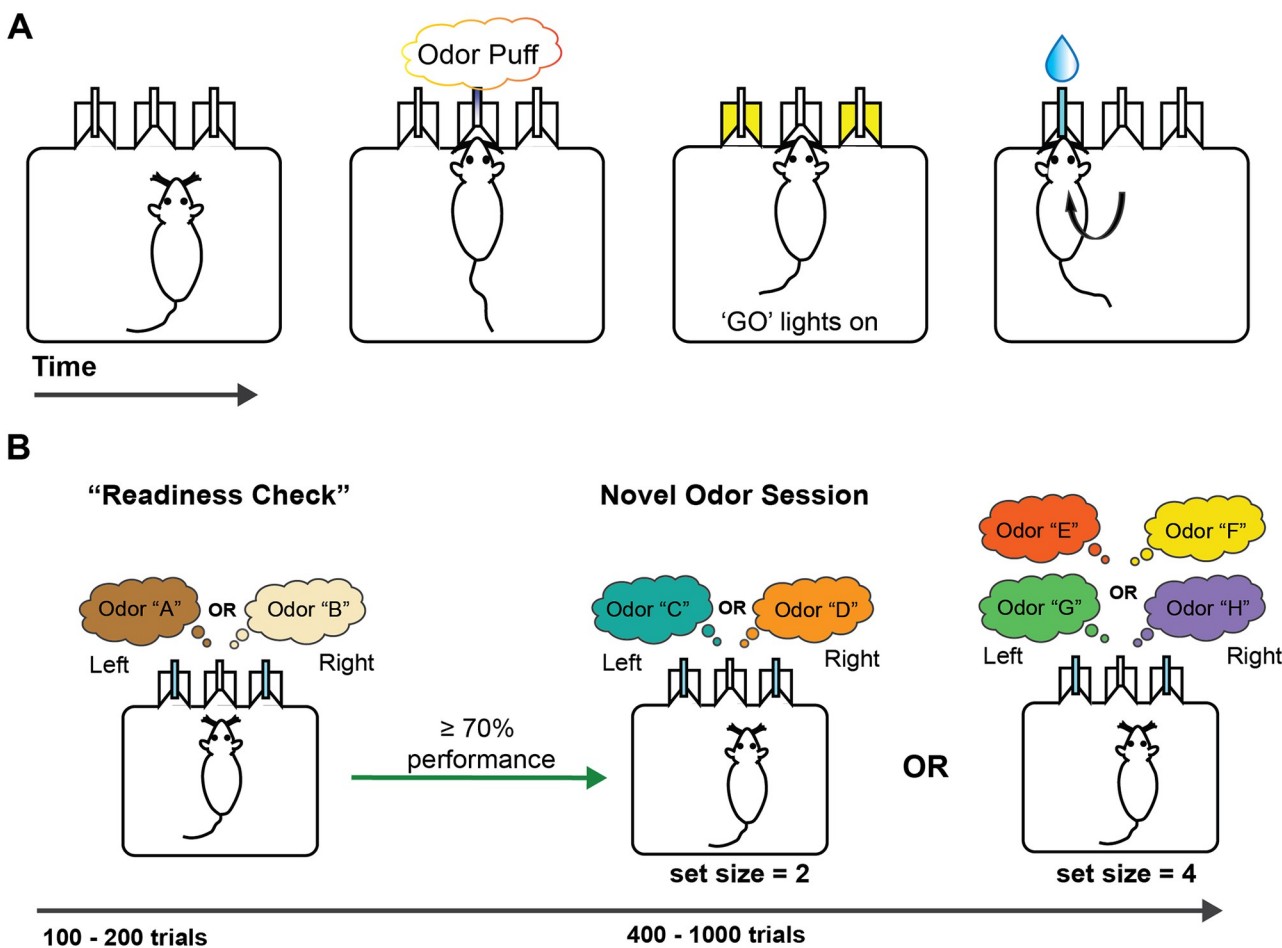

**Fig 1. Behavioral schematic of RL+WM task.** A) In the operant chamber, mice initiate a trial via nosepoke to the center port where they receive a puff of a single odorant at a time. "GO" lights in the two peripheral ports indicate the availability of water, which mice receive only if they make a correct choice. B) In each session of the full RL+WM task (following early learning), mice must reach 70% fraction correct or higher in odors A & B within the first 100 to 200 trials. If mice pass the readiness criterion, they are then exposed to novel odors (see S1 Table for a full list of odors used) in set sizes of either 2 or 4 odors (presented individually in pseudorandom order). If mice fail to pass the readiness criterion, they will retrain on odors A & B during that session.

in both set sizes (Fig 2B and 2E: Wilcoxon signed rank test: male: $W = -18971$, $p < .0001$ for set size = 2 & $W = -993$, $p < .0001$ for set size = 4; female: $W = -630$, $p = 0.017$ for set size = 2 & $W = -606$, $p < .001$ for set size = 4).

Overall learning in set size = 2 was not significantly different from set size = 4 in female mice (Fig 2D, RM 2-way ANOVA, set size: $F(1, 73) = 0.21$, $p = 0.64$), but there was a significant interaction in favor of set size = 4 between set size and time (set size * time interaction: $F(7, 512) = 2.84$, $p = 0.006$). Unlike males, females had no set size differences in performance in repeat or non-repeated trials across set sizes (t-test, repeated: $t = 1.5$, $df = 100$, $p = 0.13$; non-repeated: $t = 0.88$, $df = 100$, $p = 0.37$, Fig 2E).

Overall performance on the task (measured by fraction correct across all odor stimuli for a single session) was also comparable across all adolescent and adult ages tested (mixed-effect linear regression, male: $\beta_{age} = -0.001$, 95% CI = $[-0.002, 0]$, $p = 0.207$; female: $\beta_{age} = -0.001$, 95% CI = $[-0.003, 0.002]$, $p = 0.514$, Fig 2C and 2F).

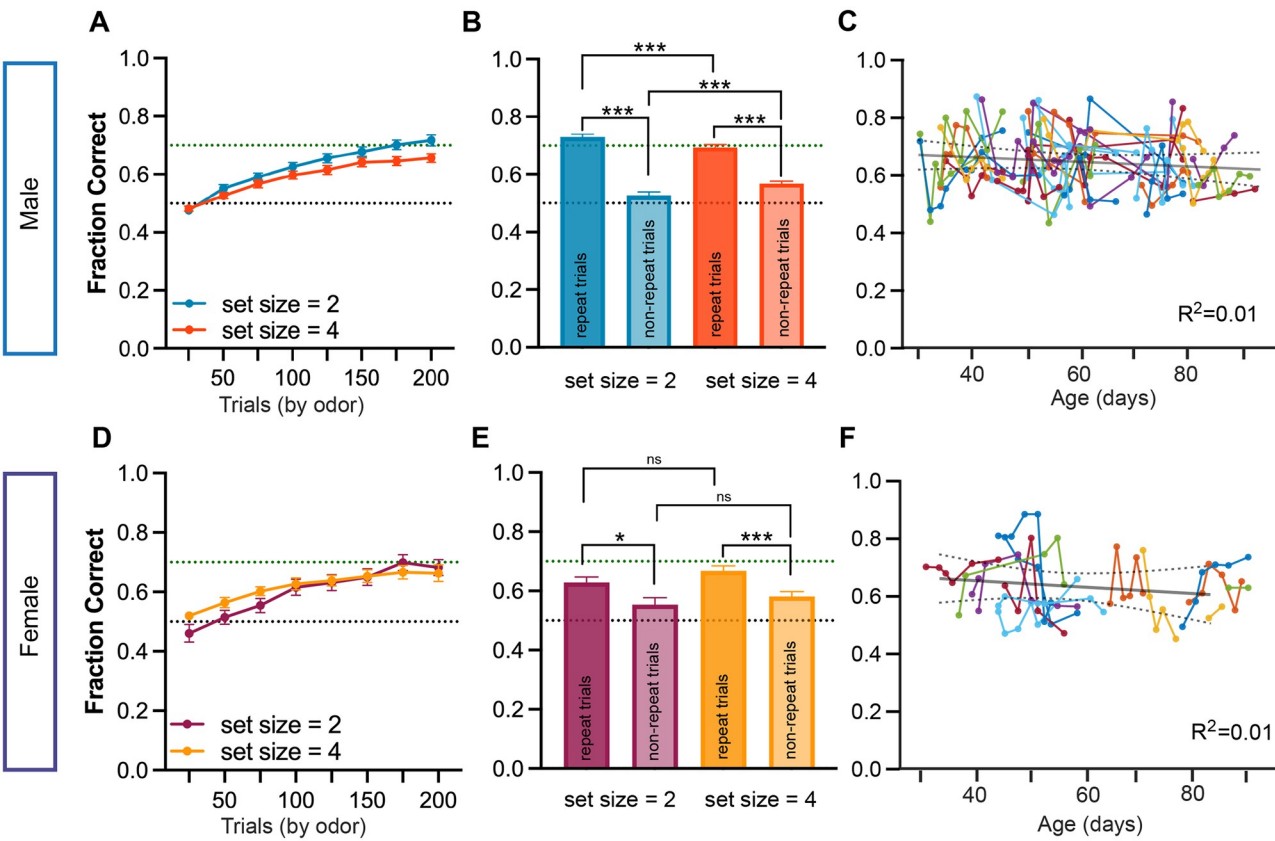

**Fig 2. Male and female mice learn RL+WM behavior task similarly across set sizes and development.** Within the first 200 trials per odor in the novel odor phase, both males (A) and females (D) showed an average performance (fraction correct) significantly above chance (dotted black line at 50%, dotted green line at 70%) in both set size = 2 and set size = 4. Trials are binned as trials per stimulus to allow for direct comparison across set sizes. Males showed a small but significant difference in performance between set sizes that grew more pronounced near the end of each session (set size: $p = 0.04$, set size * time interaction: $p = 0.01$). Female mice showed a significant interaction between set size and time that favored set size = 4 (set size * time interaction: $p = 0.006$). (B, E) Performance in males and females showed significant benefit when an odor stimulus was repeated two trials in a row when compared to non-repeated stimulus trials (males: $p < 0.0001$ in both set sizes; females: $p = 0.01$ for set size = 2 and $p < 0.001$ for set size = 4). This effect was observed in both set sizes for both sexes. However, in male mice, while performance in repeat trials was higher in set size = 2 than in set size = 4 (possibly driven by a higher proportion of repeat trials when mice are exposed to 2 odors compared to 4), this relationship was reversed when performance in non-repeat trials was examined ($p < 0.0001$). (C,F) Mean fraction correct in males and females did not change in sessions across development P30–90, a period that includes both adolescent development and early young adulthood in mice. Lines connect up to three sessions per mouse and show performance for first 200 trials per odor in both set size = 2 and set size = 4 for all sessions analyzed, with dotted lines reflecting 95% confidence intervals. Error bars indicate SEM. Asterisks indicate the result of RM 2-way ANOVA (A,D) and either a paired t-test for within session comparisons (repeat with non-repeat in the same set size) or an unpaired t-test and Wilcoxon signed rank test (B,E). *** $p < 0.001$, ** $p < 0.01$, * $p < 0.05$.

Based on the minimal effect of set size in mice, which could be primarily driven by repeat trials, we inferred that learning in both set sizes was most likely supported by the RL learning system, as a human-like efficient working-memory component would have elicited strong set size effects, including in non-repeat trials. Furthermore, unlike the gradual emergence of a set size difference that we observed in male mice, human-like working memory would be expected to contribute most to performance early in learning and favor smaller set sizes.

## Logistic regression models

To build from our behavioral results and to better characterize the effect of recent history separately from the effect of long-term cumulative reinforcement, we next analyzed trial-by-trial

choices for each individual session in a series of logistic regressions. Our first regression (regression #1), was simple with correct history, 1-back repeat (odor stimulus), and 2-back repeat as predictors (see Materials and methods). We found for set size = 2 that there was a significant positive effect of correct history across sessions for both males and females (Fig 3A and 3E, t-test, males: $t = 14.51$, $df = 31$, $p < 0.0001$, females: $t = 14.46$, $df = 14$, $p < 0.0001$), confirming that mice were more likely to make a correct choice the more they experienced reward, as expected from an incremental RL process. The regression coefficient for 1-back repeated odor cue, *1-back*, was also significantly above 0 for males, but not for females (one sample *t*-test, males: $t = 4.99$, $df = 31$, $p < 0.0001$, females: $t = 1.03$, $df = 14$, $p = 0.31$) indicating a possible sex difference in weighting a previous trial's outcome. There was no 2-trial back effect for either sex (one sample *t*-test, males: $t = 0.47$, $df = 31$, $p = 0.63$, females: $t = 0.99$, $df = 14$, $p = 0.33$). Together, these results parallel the weak behavioral effect we found in females for repeated cues (Fig 2E) and indicate that male mice might be maintaining and leveraging information in the immediately previous trial, $t - 1$, but not further in the past. The significantly higher *1-back* coefficient for male mice compared to females (Unpaired t-test, $t = 2.09$, $df = 45$, $p = 0.04$), prompted us to continue to analyze male and female data separately in subsequent analyses. Importantly, all regressors that significantly captured male mouse behavior in set size = 2, were also significant in set size = 4 and there was only a single difference between set sizes in female mice (see Fig 3). This replicates our behavioral finding that mice in the RL +WM task perform similarly in set size = 2 and set size = 4 sessions.

To further probe which information in the previous trial was used, we examined additional regressions by categorizing trial characteristics such as repeating cues (1-back repeat = 1) vs. non-repeating cues (1-back repeat = 0), and previously rewarded (reward = 1) vs. non-

Regression # 1:   $p(\text{correct}) = \text{logit}^{-1}(\beta_0 + \beta_1 * [\text{correct-history}] + \beta_2 * [\text{1-trial-back}] + \beta_3 * [\text{2-trial-back}])$

Regression # 3:   $p(\text{correct}) = \text{logit}^{-1}(\beta_0 + \beta_1 * [\text{correct-history}] + \beta_2 * [\text{main-effect-reward}] + \beta_3 * [\text{reward-repeat}] + \beta_4 * [\text{noreward-repeat}])$

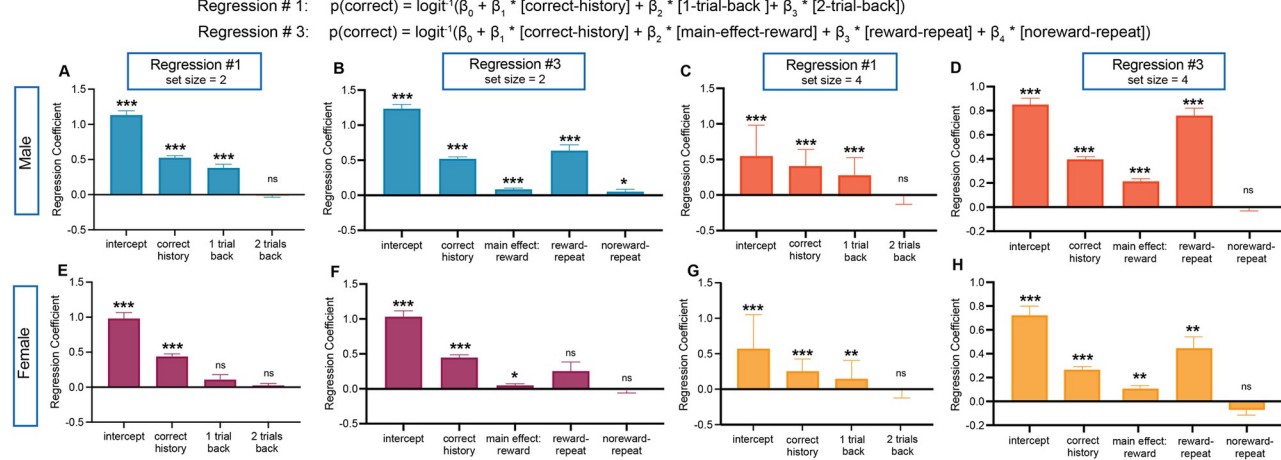

**Fig 3. Summary regression coefficients from main regressions run in set size = 2 and set size = 4.** In order to understand the influence of past trials on current trial *t*, we developed and tested multiple regressions with a logit link function and fit them to individual sessions. Here we show our preliminary regression, regression #1, and our winning regression model (regression #3) after testing a total of six regressions (see S1 Fig). All regressions included a theoretical marker for RL (correct history) and a series of different one-back trial identities that could be used to approximate WM. The influence of correct history for both male and female mice across both set sizes and regressions was significant, indicating the use of RL. By looking at one-back and two-back trials in regression #1, we found that one-back was significantly above 0 for male mice in both set sizes and for females in set size = 4 only ($p = 0.04$). We also found that two-back was not significant for either sex. For males in regression #3 for both set size = 2 and set size = 4, all coefficients were significantly above 0. For females, there was only a significant effect of reward for both set size = 2 and set size = 4 and in the interaction between reward and repeat in set size = 4 only. Normality tests on grouped regression coefficients from individual sessions were run as described in Materials and methods and parametric (one sample *t*) or non-parametric (Wilcoxon signed rank) tests were applied accordingly. Each column reflects the mean of individual sessions (3 from each animal in each set size) and error bars correspond to SEM. **** $p < 0.0001$ *** $p < 0.001$, ** $p < 0.01$, * $p < 0.05$.

rewarded (reward = 0) trials with a series of different combinations and interactions (see Materials and methods). Using information criterion tests we determined that both regression #2 and regression #3 had the lowest scores (S1(A) Fig). We chose regression #3 to describe below and outline the results from regression #2 in the supporting information (see S4–S6 Figs).

In regression #3, we looked at the effect of correct history (an identical regressor to regression #1 that measured the influence of reward history on current outcome), previous reward (whether the previous choice was rewarded or not), and repeat odor cues in previously rewarded and non-rewarded trials. We found that all regression weights for set size = 2 were positive (Fig 3B and 3F) and that there was a significant effect of correct history for both sexes (males: $t = 14.78$, $df = 31$, $p < 0.0001$. Females: $t = 14.61$, $df = 14$, $p < 0.0001$), indicating again that mice are using some form of RL to guide choices. Previously rewarded trials strongly influenced the outcome on trial $t$ for males, but not for females (males: $t = 4.89$, $df = 31$, $p < 0.0001$; females: $t = 1.03$, $df = 14$, $p = 0.32$). Male mice also were strongly influenced by the odor stimulus repeating only when the previous outcome was rewarded ($t$-test and Wilcoxon signed rank test, males, reward-repeat: $t = 5.2$, $df = 31$, $p < 0.0001$; females, reward-repeat: $W = 24$, $p = 0.52$; males, noreward-repeat $t = 1.28$, $df = 31$, $p = 0.20$, t-test, females, noreward-repeat: $t = 0.33$, $df = 14$, $p = 0.74$). These results suggest that male mice might be able to leverage memory from the last trial, confirming the conclusion of our initial simple regression that some form of adaptive one-back information is used. These regressors suggest an additional strategy to RL where mice are not simply updating action values based on rewarded outcomes, but also paying attention to stimulus-action contingencies in one-back trials, as revealed by the stronger influence of reward-repeat than simply rewarded trials (Fig 3B, paired $t$-test, $t = 4.54$, $df = 31$, $p < 0.0001$). Similar results in set size = 4 were seen for both male and female animals, except for the slightly less positive effect for males and females of repeated stimuli in non-rewarded trials (Fig 3D and 3H), possibly driven by the lower proportion of repeat trials expected in set size = 4 as we saw in behavior.

To investigate potential changes in an animal's trial-by-trial behavior throughout its development, we employed mixed-effects models. In these models, each session's regression coefficient served as the dependent variable, while session age was used as the predictor. This approach allowed us to examine how specific aspects of behavior, captured by the regression coefficient weights, varied across different developmental stages and enabled us to statistically account for repeated sessions by the same animal (see Materials & methods). Most significant developmental changes were seen in male, not female, animals (see Figs 4 and 5). In regression #3 the main effect of reward was insignificant for both sexes (males: Fig 4C, $\beta_{age} = 0.001$, 95% CI = [−0.001, 0.003], $p = 0.28$; females: Fig 5C, $\beta_{age} = 0.00$, = 95% CI = [−0.004, 0.005], $p = 0.82$). For males, the regressors that captured the influence of repeating stimuli regardless of reward outcome, both increased over development (Fig 4D and 4E: $\beta_{age} = 0.03$, 95% CI = [0.022, 0.038], $p < 0.0001$; $\beta_{age} = 0.009$, 95% CI = [0.004, 0.013], $p < 0.0001$). Interestingly, the influence of correct history did not change over development in either males or females. This, combined with the developmental stability of the main effect of reward, together indicated that the contribution of RL-like learning in this simple associative learning task context is consistent across adolescence into early adulthood. There were some differences in findings across set sizes, but major effects (repeat-reward for males and a lack of significance for any coefficient for females) were replicated in set size = 4.

## Computational modeling

To build on our regression results and obtain a more quantitative and mechanistic understanding of potential one-back effects we next used computational modeling. The models we

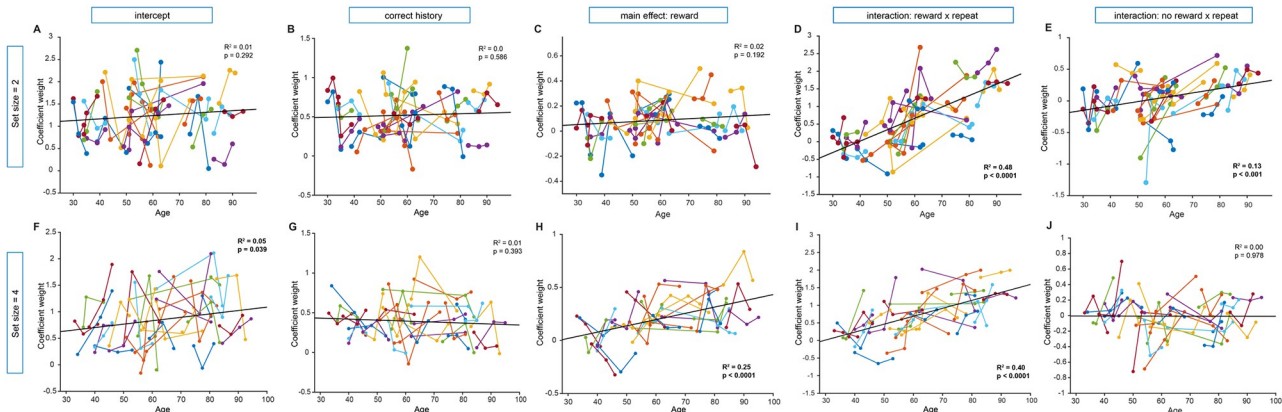

**Fig 4. Male regression coefficients for regression #3 across development in set size = 2 and set size = 4.** In order to understand the relationship between age and the predictors of current choice in trial $t$ (see Fig 3 for the logistic regression summary), we looked at whether coefficient weight (y-axis) would change over development (x-axis) for male mice. It is possible that as mice age, they rely to differing extents on separable learning systems (for example, RL, captured by 'correct history' and, short-term memory, captured by one-back combinations of stimuli and reward). For each coefficient from regression #3, our winning regression (see S1 Fig), we report the $R^2$ and $p$-value from a simple linear regression (best fit line) and the results of a mixed linear model in the main text. While 'correct history,' a measurement of RL, was significantly above 0 for both set sizes (Fig 3), the coefficient weight did not change across age. (B,G). The intercept for both set sizes did not change significantly across age (A,F) and some developmental changes that were significant in set size = 2 (reward-repeat: $p < 0.0001$; noreward-repeat: $p < 0.0001$) were also reflected in set size = 4 (reward-repeat: $p < 0.0001$; main effect of reward: $p < 0.0001$). There was a difference between the regression weight significance across set sizes for noreward-repeat and main effect of reward (C,J), which could have been influenced by the addition of two more stimuli.

compared were generally based on RL tracking of stimulus-action values, but also included dependency on previous trials for choice policy (see Materials and methods). Using model comparison we identified a complex model with multiple parameterized one-back policies as the best fitting and most recoverable model (Fig 6B). The winning model contained a single learning rate $\alpha$ and noise parameter $\beta$, with learning from negative outcomes $\alpha_- = 0$; as well as three free parameters characterizing one-back policy bias as a function of last trial's

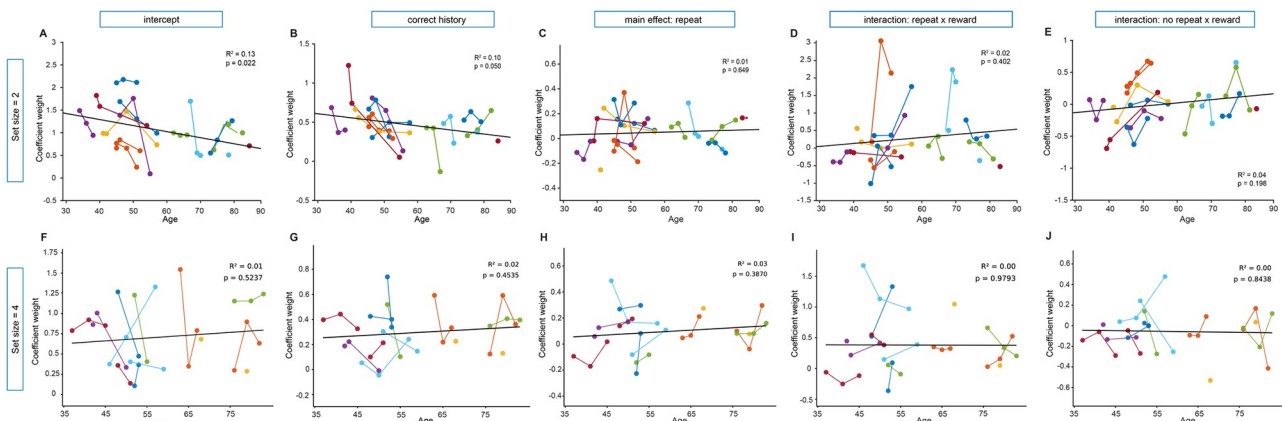

**Fig 5. Female regression coefficients for regression #3 across development in set size = 2 and set size = 4.** The influence of age on regression coefficients from our winning regression, regression #3, for both set size = 2 and set size = 4 are formatted in the same way as males in Fig 4. All statistics are done identically and are reported in the text. There was a slight decrease in the regression coefficient weight of intercept for set size = 2 (A), but not set size = 4 (F). Outside of this, there were no significant changes across development in set size = 4, but there was a significant change in reward-repeat ($p < 0.0001$) and noreward-repeat ($p < 0.0001$) only in set size 2. These results are in contrast to significant changes seen across development in male mice in regression #3 that had some consistencies across set sizes (Fig 4).

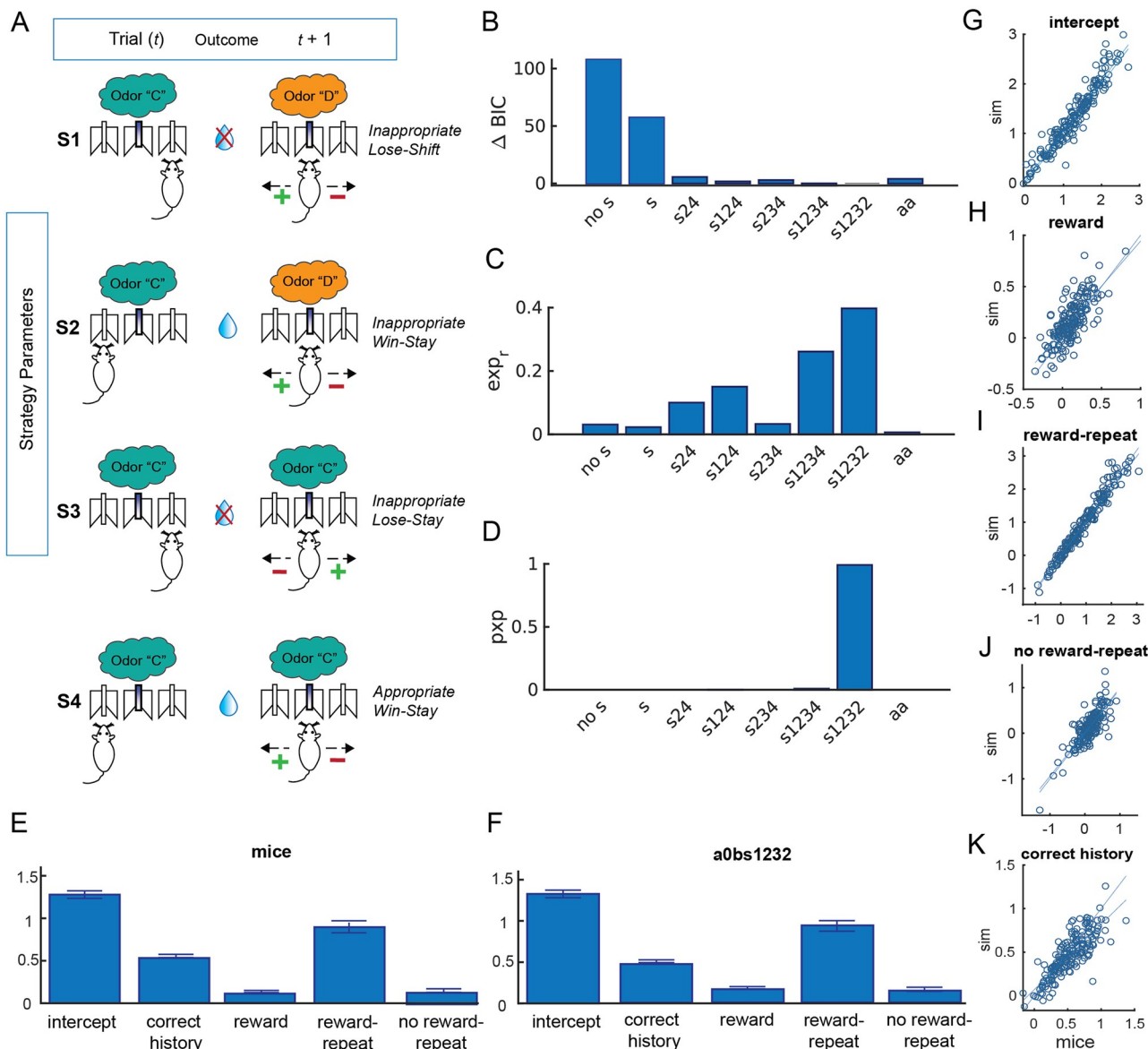

**Fig 6. Model comparison and validation.** (A) We examined 4 parameters that isolated different ways mice might use 1-trial back information (S1-S4). Later in the best-fitting model, S2 and S4 were collapsed to one parameter to reduce complexity. Each left cartoon indicates the stimulus and the mouse's action at time $t$ and the cartoon on the right reflects the choice options at time $t$+1 that a mouse has when presented with either stimulus. The plus/minus sign reflect the positive/negative direction of the specific strategy parameter and the name of each strategy parameters describes the positive value of the parameter at time $t$+1. (B) The winning model a0bs1232 (where a indicates a free $\alpha_+$ parameter, 0 indicates that $\alpha_- = 0$ is fixed, and s1232 indicates three free S parameters with S4 = S2) was compared to other a0b models with various combinations of strategy parameters (s) or no strategy parameters (nos) as well as a model leaving the negative learning rate free (aabs1232). All competing models built from basic RL model described in Materials and methods. The winning model had an average BIC of 859.4, not significantly different from the full model a0bs1234 (BIC = 857.6191). However, a Bayesian model comparison analysis showed that a0bs1232 was significantly more frequent across sessions (C; posterior frequency 39.74% vs. 26.11% for a0bs1234), with a protected exceedance probability (pxp, [12]) of 0.991 (D), confirming that it was the best model at the group level. (E-F) Example logistic regression. To validate our winning model we analyzed simulated behavior from parameters fit on each individual and session with the same logistic regression approach. We found that the simulated behavior captured well the impact of repeat, reward, their interaction, and the effect of reward history on a session by session basis, as shown by the group level effect (E-F) and tight correlations between mouse behavior and simulated behavior using fit parameters for regression #3 (sim; G-K).

characteristics (see Fig 6A for parameter schematic). These three one-back policy parameters emerged from an initial group of four parameters: two in response to non-rewarded outcomes (S1 and S3) and two in response to rewarded outcomes (S2 and S4). Each parameter is named for action associated with the positive direction (Fig 6A). The first, parameter S1, or "Inappropriate Lose-Shift," biased against repeating the previous trial's choice when the stimulus changed and the animal's previous choice was unrewarded. Parameter S3, or "Inappropriate Lose-Stay," biased policy towards repeating the previous choice even when the stimulus stayed the same and the animal's previous choice was unrewarded (repeating an incorrect choice), thus capturing perseveration. We initially considered separate S2 and S4 parameters (see Fig 6A): "Inappropriate Win-Stay," S2, captured staying with the previous trial's choice after a rewarded trial even when the stimulus changed; and "Appropriate Win-Stay," S4, indicated staying with the previous trial's choice that was rewarded when the stimulus stayed the same. But, because S2 and S4 were highly correlated with each other, we reduced model complexity by setting S2 = S4, a parameter we called "Stimulus Insensitive Win-Stay". This parameter, biased towards repeating the previous trial's choice after a win independently of whether the past stimulus changed or stayed the same.

We explored other models (including various parameterizations of last-trial-dependent policy biases), and found that this 5-parameter model was best able to recover the data (see Figs 3 and 6E–6K, Materials and methods). We also confirmed that model parameters were identifiable (See Materials and methods, S2 Fig) and that the model captured individual differences across mice and sessions accurately (Fig 6G–6K).

Once the data from each mouse and each session were fit using the winning model, we examined how the fit model parameters changed with age when mice were learning novel odor sets. The average learning rate $\alpha_+$ and softmax noise parameter $\beta$ in set size = 2 and set size = 4 were stable across this period of development in both sexes (Figs 7A, 7B, 7F, 7G, 8A, 8B, 8F and 8G) (mixed linear model, males: $\alpha_+$ parameter, set size = 2: $\beta_{age} = -0.004$, 95% CI = [−0.025, 0.017], $p = 0.716$; $\beta$ parameter, set size = 2: $\beta_{age} = 0.002$, 95% CI = [−0.006, 0.011],

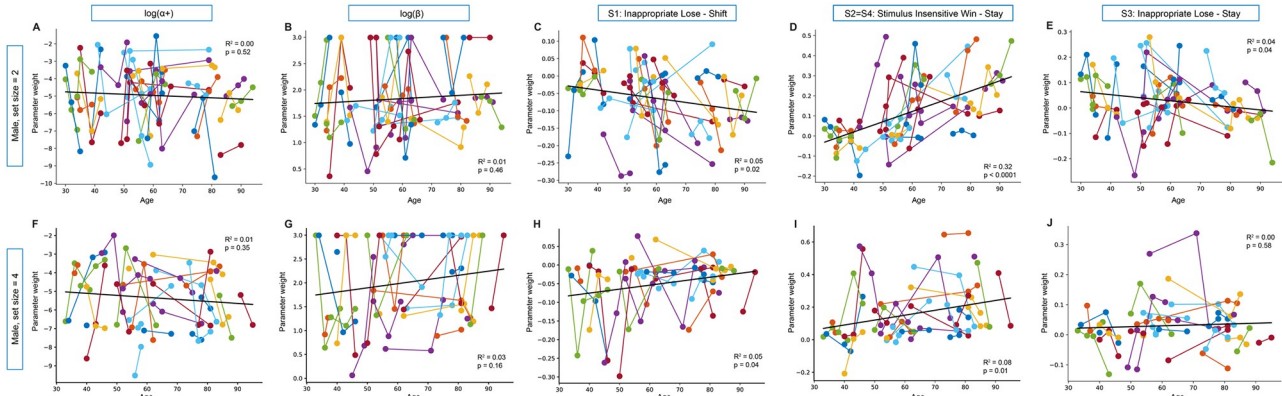

**Fig 7. Strategy parameters across development for male mice in set size = 2 and set size = 4 from winning computational model.** In order to understand the relationship between age and the parameters from our winning model (see Fig 6 and Materials and methods for model details), we looked at whether parameter weight (y-axis) would change over development (x-axis) for male mice in set size = 2 (A-E) and set size = 4 (F-J). Parameter values are colored by individual with separate sessions connected by lines. For each parameter, we report the $R^2$ and p-value from a simple linear regression (best fit line) on the figure and the results of a mixed linear model with age as the predictor variable and the parameter as the dependent variable in order to better account for variability driven by repeat sessions by individual mice is reported in the main text (see Materials & methods for more information about the analyses). RL parameter $\alpha_+$ learning rate and decision noise parameter softmax $\beta$ were stable across development in both set size = 2 (A-B) and set size = 4 (F-G). Both parameters S1 "Inappropriate Lose-Shift" and S3 "Inappropriate Lose-Stay" decreased significantly with age in set size = 2 (C: $p = 0.02$, E: $p = 0.04$), but not in set size = 4 (H,J). However, strategy parameter, S2 = S4 "Stimulus Insensitive Win Stay," increased significantly for male mice in both set size = 2 (D:$p < 0.0001$) and set size = 4 (I: $p = 0.03$).

$p = 0.637$; males $\alpha_+$ parameter, set size = 4: $\beta_{age} = -0.009$, 95% CI = [−0.033, 0.015], $p = 0.450$; $\beta$ parameter, set size = 4: $\beta_{age} = 0.007$, 95% CI = [−0.005, 0.020], $p = 0.251$; females $\alpha_+$ parameter, set size = 2: $\beta_{age} = -0.016$, 95% CI = [−0.066, 0.035], $p = 0.546$; $\beta$ parameter, set size = 2: $\beta_{age} = 0.002$, 95% CI = [−0.02, 0.02], $p = 0.858$, females $\alpha_+$ parameter, set size = 4: $\beta_{age} = 0.012$, 95% CI = [−0.064, 0.088], $p = 0.753$; $\beta$ parameter, set size = 4: $\beta_{age} = 0.009$, 95% CI = [−0.032, 0.050], $p = 0.656$).

The one-back strategy parameter that reflects perseveration after a rewarded trial, parameter S2 = S4: "Stimulus Insensitive Win-Stay," grew with age in males in set size = 2 (Fig 7D, $\beta_{age} = 0.006$, 95% CI = [0.004, 0.007], $p < 0.0001$), and was replicated in set size = 4 (Fig 7I, $\beta_{age} = 0.003$, 95% CI = [0.00, 0.006], $p = 0.032$). Growth with age in the S2 = S4 parameter was not seen in females in either set size (Fig 8D, set size = 2:$\beta_{age} = 0.001$, 95% CI = [−0.004, 0.005], $p = 0.713$; Fig 8I, set size = 4: $\beta_{age} = 0.001$, 95% CI = [−0.007, 0.010], $p = 0.766$).

The one-back strategy parameters S1, "Inappropriate Lose-Shift", and S3, "Inappropriate Lose-Stay" indicate how an animal responds to a non-rewarded trial. Both of these parameters showed inconsistent age related changes between sexes and set sizes, which parallels the low and inconsistent influence of non-rewarded repeat trials in regression #3. Male animals in set size = 2 had a significant decrease across age for parameter S1 (Fig 7C, $\beta_{age} = -0.002$, 95% CI = [−0.003, 0.0], $p = 0.023$), but not in set size = 4 (Fig 7H, $\beta_{age} = 0.00$, 95% CI = [−0.001, 0.002], $p = 0.474$). For parameter S3, male animals in set size = 2 also had a significant decrease across development (Fig 7E, $\beta_{age} = -0.001$, 95% CI = [−0.003, 0.0], $p = 0.04$), but not in set size = 4 (Fig 7J, $\beta_{age} = 0.00$, 95% CI = [−0.002, 0.001], $p = 0.607$). We discounted these changes due to lack of consistency.

Female animals showed decreases in S1 with age that were at trend level in set size = 2 and significant in set size = 4(females S1, set size = 2, Fig 8C: $\beta_{age} = -0.002$, 95% CI = [−0.004, 0.00], $p = 0.08$; Fig 8H, set size = 4: $\beta_{age} = -0.003$, 95% CI = [−0.006, 0.00], $p = 0.033$). In females the S3 parameter showed no change with age in either set size (females, S3, set size = 2, Fig 8E: $\beta_{age} = 0.00$, 95% CI = [−0.003, 0.002], $p = 0.74$; Fig 8J, set size = 4: $\beta_{age} = 0.001$, 95% CI = [−0.002, 0.004], $p = 0.556$).

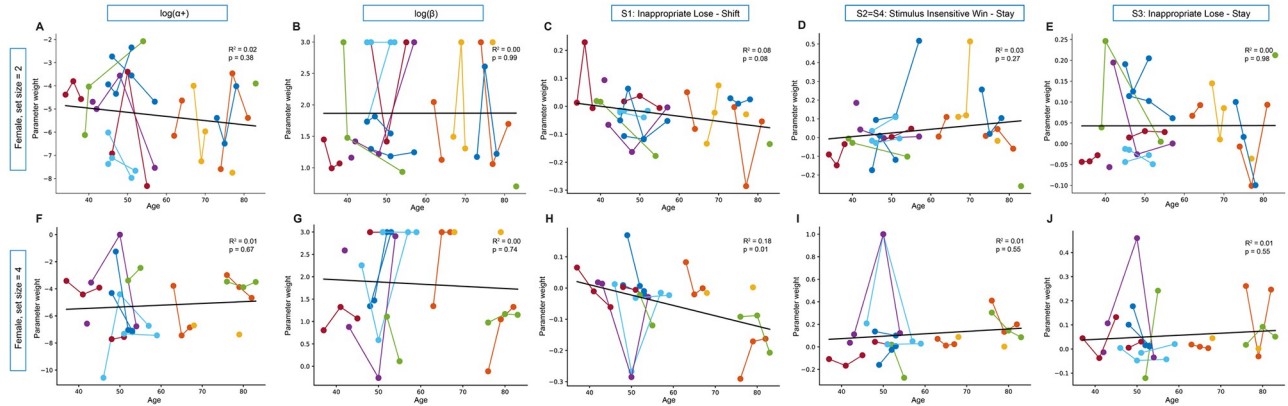

**Fig 8. Strategy parameters across development for female mice in set size = 2 and set size = 4 from winning computational model.** Parameters from our winning model across age for female animals are formatted in the same way as Fig 7 and all mixed linear model statistics are reported in the main text. RL parameter $\alpha_+$ learning rate and decision noise parameter softmax $\beta$ were stable across development in both set size = 2 (A-B) and set size = 4(F-G). Both parameters S2 = S4 "Stimulus Insensitive Win Stay" and parameter S3 "Inappropriate Lose-Stay" did not change across development in either set size = 2 (D,I) or set size = 4 (E,J). However, female mice had a significant decrease in parameter S1 "Inappropriate Lose-Shift" in set size = 4 (H: $p = 0.04$) with a trend in a similar direction in set size = 2 (C: $p = 0.08$).

We next looked for effects of session (meta-learning from repeat exposure) for each of the five parameters. Set size = 2 and set size = 4 sessions were interleaved, so we combined and analyzed session data from set size = 2 and set size = 4 in the order mice experienced them. Both $\alpha$ learning rate and softmax noise parameter $\beta$ did not show an effect of session for either sex (Fig 9A, 9B, 9F and 9G: males, $\alpha_+$: $\beta_{session}$ = −0.094, 95% CI = [−0.246, 0.058], $p$ = 0.22; males, $beta$: $\beta_{session}$ = 0.053, 95% CI = [−0.023, 0.128], $p$ = 0.17; females, $alpha_+$: $\beta_{session}$ = −0.022, 95% CI = [−0.304, 0.261], $p$ = 0.88; females, $beta$: $\beta_{session}$ = 0.040, 95% CI = [−0.089, 0.168], $p$ = 0.54). This is consistent with what we saw in behavior (Fig 2C and 2F) and in the correct history parameters in our regression analyses (Figs 4, 5A and 5F). However, use of the three one-back parameters changed significantly across sessions in males with S1 "Inappropriate Lose Shift" decreasing across sessions ($\beta_{session}$ = −0.010, 95% CI = [−0.018, −0.002], $p$ = 0.01), S2 = S4 "Stimulus Insensitive Win Stay" increasing ($\beta_{session}$ = 0.018, 95% CI = [0.005, 0.032], $p$ = 0.009), and S3 "Inappropriate Lose Stay" decreasing ($\beta_{session}$ = −0.018, 95% CI =

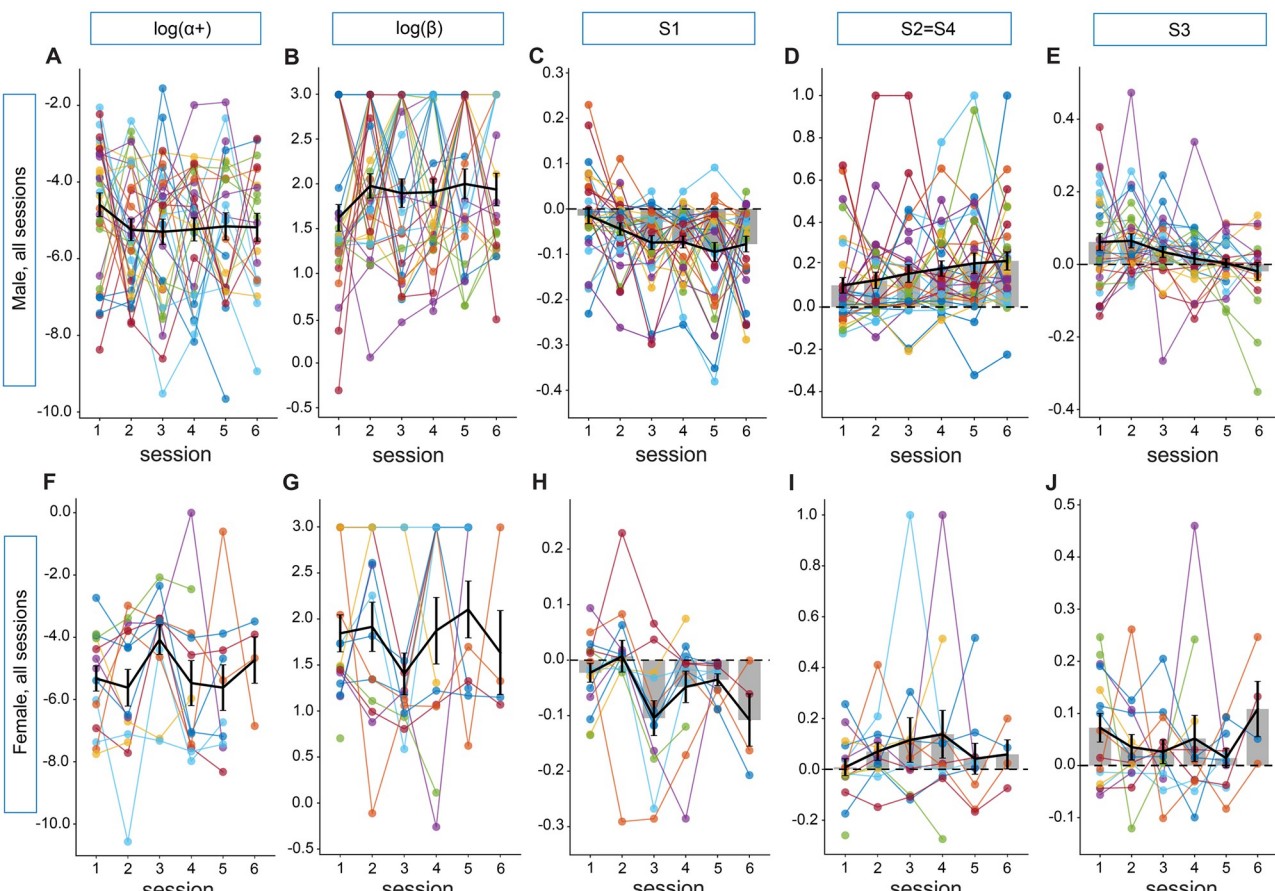

**Fig 9. Effect of session on winning model parameters for set size = 2 and set size = 4 for both male and females.** To test if mice adjusted one-back strategies with experience, we next compared how parameter weights (y-axis) changed across the 6 sessions analyzed for each mouse (x-axis). Since set size = 2 and set size = 4 days were interspersed, we combined set size data and analyzed sessions chronologically. Since each mouse had multiple sessions, each individual was colored and their sessions connected as described previously (Fig 7). Gray bars indicate mean values for each session with SEM and a line connecting each mean for better visualization. Dotted lines are set to 0 for strategy parameters. Similar to what was seen in comparisons between regression coefficients and age, there was no effect of experience on $\alpha_+$ (A,F) or $\beta$ (B,G) for either sex. Use of one-back strategy parameters changed significantly across sessions for male mice with (C) S1 "Inappropriate Lose Shift" decreasing across sessions, $p$ = 0.01 (D) S2 = S4 "Stimulus Insensitive Win Stay" increasing, $p$ = 0.009 and (E) S3 "Inappropriate Lose Stay" decreasing, $p$ < 0.0001. Female mice showed a trend in the same direction of male mice in (H) S1 "Inappropriate Lose Shift", $p$ = 0.06, but did not show any experience related changes in (I) S2 = S4 "Stimulus Insensitive Win Stay" or (J) S3 "Inappropriate Lose Stay". Full statistics from mixed linear models are reported in the main text.

[−0.027, −0.009], $p < 0.0001$). These experience-dependent trends are in the same direction as male set size = 2 changes over age (Fig 7C–7E), although this trend is less captured by set size = 4 (Fig 7H–7J). Together, the opposing trends in S1 and S2 = S4 could indicate that male mice are more likely to stick with their previous choice, regardless of reward or stimuli, as they gain experience in the task (see Fig 6A). A decrease in S3 across sessions shows that males also switch their response following an unrewarded trial when the stimuli stays the same, which may reveal an increase in task knowledge over time. Females, showed a trend toward a decrease in S1 parameter weights over sessions, similar to males, but no significant changes across sessions in one-back strategies (Fig 9H–9J), S1: $\beta_{session} = −0.012$, 95% CI = [−0.026, 0.001], $p = 0.068$; S2 = S4: $\beta_{session} = 0.013$, 95% CI = [−0.019, 0.044], $p = 0.42$; S3: $\beta_{session} = −0.003$, 95% CI = [−0.018, 0.013], $p = 0.75$).

These modeling results suggest that mice have access to, and leverage information from, short-term memory for making choices in learning situations as humans do. However, they use this information for short-term one-back choice strategies instead of human-like working memory.

## Adult behavior after pre-pubertal gonadectomy

Males showed significant changes in one back strategies with age and experience. To test if these changes were driven by increases in gonadal hormones after puberty in males, we performed gonadectomy (GDX) or sham surgeries prior to the onset of puberty (P25) in a separate cohort of mice (see Materials and methods for additional information). GDX and sham mice were trained in the task as groups described above at young adult ages (P61-P96) when puberty is complete. We found that both groups learned comparably across the first 200 trials per odor stimulus (Fig 10A) (2-way RM ANOVA: condition: $F(1, 82) = 0.57$, $p = 0.44$; time: $F(3.8, 311.9) = 49.06$; condition x time: $F(7, 574) = 0.2955$, $p = 0.95$). We next fit GDX and sham animals to our same regression models as described previously (see Materials and methods, see S1 Fig) and examined differences between GDX and sham (S7(A) and S7(B) Fig). We found that GDX and sham males were comparable for intercept and correct history in both regressions. The two groups differed significantly in one-back trial in regression #1 (Unpaired $t$-test: $t = 3.63$, $df = 9$, $p = 0.005$) and both reward-repeat and no reward-repeat in regression #3 (Unpaired $t$-tests, reward-repeat: $t = 2.97$, $df = 9$, $p = 0.01$; no reward-repeat: $t = 2.48$, $df = 9$, $p = 0.03$) (see S7(A) Fig). We next fit multiple models and the winning computational model

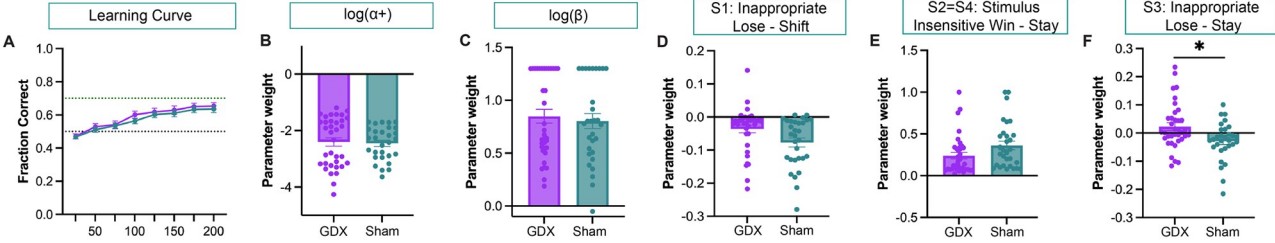

**Fig 10. Age-matched GDX and sham controls show no differences in behavior or RL learning parameters and minimal differences in strategy parameters.** A separate cohort of male mice underwent gonadectomy (GDX) (n = 6) or sham (n = 5) surgery prior to puberty to test if gonadal testosterone at puberty was required for age-dependent changes in males. Mice were tested on behavior between P61 and P96. Three sessions from set size = 2 and set size = 4 data were pooled and mixed linear models were used to compare groups while controlling for repeated measures. (A) GDX and sham animals showed comparable learning curves. (B) $\alpha_+$ learning rate was comparable between groups. (C) decision noise parameter $\beta$ was comparable between groups. There were also no differences across (D) S1: Inappropriate Lose-Shift and (E) S2 = S4: Stimulus-Insensitive Win-Stay. However, there was a difference between GDX and sham in (F) S3: Inappropriate Lose-Stay with sham animals having lower values than GDX. Notably, sham animals were also significantly lower than intact mice of the same age range (S7(K) Fig $\beta_{conditon} = −0.15$, 95% CI = [−0.20, −0.08], $p < 0.0001$), suggesting there may be an effect of sham surgery on this metric or variation in cohorts.

was the same best fit model as above for intact mice (posterior frequency: 48.2% vs. 18.4% for the second highest; protected exceedance probability: 1.0). We validated the model fit in both overall (S7(C) Fig) and trial-by-trial (S7(I) and S7(J) Fig) learning. We also found that the model parameters were recoverable (S7(D)–S7(H) Fig). When we compared model parameters between groups, we found that there were no differences for $\alpha_+$ learning rate (Fig 10B, $\beta_{condition}$ = −0.88, 95% CI = [−2.09, 0.33], $p$ = 0.15) or the decision noise parameter $\beta$ (Fig 10C, $\beta_{condition}$ = 0.05, 95% CI = [−0.34, 0.46], $p$ = 0.77). There were no differences across S1: Inappropriate Lose-Shift or S2 = S4: Stimulus-Insensitive Win-Stay, both parameters that showed age and experience effects in intact male mice (Fig 10E, S1: $\beta_{condition}$ = 0.009, 95% CI = [−0.03, 0.05], $p$ = 0.39; (Fig 10F, S2 = S4: $\beta_{condition}$ = 0.07, 95% CI = [−0.04, 0.19], $p$ = 0.19. There was, however, a difference between GDX and sham in S3: Inappropriate Lose-Stay (Fig 10G, $\beta_{condition}$ = −0.14, 95% CI = [−0.20, −0.08], $p < 0.0001$). Further analysis suggested sham mice drove this difference because their values were lower than both GDX and intact male mice (S7(K) Fig). These experiments suggest that gonadal hormones in males at puberty are not driving differences we see with age in one-back strategies in intact males.

## Discussion

We tested mice in a conditional associative learning task with a set size manipulation that has been used previously to disentangle contributions of working memory (WM) from reinforcement learning (RL) systems. A goal of our study was to identify the extent to which mice use WM and the age at which both WM and RL become adult-like. We hypothesized that mice would show a signature of WM in low, not high set sizes and that reliance on WM would increase throughout development.

Mice readily learned the odor-based task when presented with both set size = 2 and set size = 4 sessions, moving gradually from chance performance to above 70% correct over the course of 200 trials per odor. While male animals performed significantly better in set size = 2 than set size = 4 (Fig 2A), this emerged later in the session and was likely driven by more repeat trials in set size = 2 than set size = 4, rather than a notable learning difference between set sizes. Gradual learning of stimulus-action associations is consistent with the use of an RL strategy [10] and this, combined with the learning similarity between set sizes, was an initial clue that mice may not be using canonical WM to solve the task. However, we also noted better performance for male animals in trials when an odor was repeated over non-repeated trials across both set sizes (Fig 2B), a difference which could either be evidence of one-trial-back WM (presenting differently than canonical or human-like WM) or consistent with a non-WM dependent one-trial-back strategy. To parse these differences in strategy, we turned to regression analyses and fit trial-by-trial RL models for both set size = 2 and set size = 4.

These regression analyses confirmed that history of rewarded trials, a marker of RL, and certain patterns of one-back repeated, rewarded, and non-rewarded trials contributed significantly to performance in both set size = 2 and set size = 4 (see Figs 3–5). Building on our regression findings with RL models, we found that along with more standard parameters like learning rate $\alpha$ and softmax $\beta$, the inclusion of parameters weighing a variety of one-back choice strategies allowed better fit and recovery of behavioral data for individual mice. Together, these data suggest mouse performance in the RL+WM task can be well explained by RL alongside simple one-back choice strategies.

While our data show that mice use one-trial-back information to influence their decision above and beyond cumulative reward, we see no evidence that this is comparable to canonical working memory observed in human learning [8]. This lack of WM use is surprising in light of other studies that show mice and rats perform spontaneous alternation or can make choices

using information that has been maintained over a delay [6, 7]. Successful performance on these other tasks suggest there is some form of working memory available to rodents by early adulthood that is not recruited in our task. This could be driven, in part, by our task design which requires rodents to connect an odor cue with an action and/or a spatial location, rather than just a requirement to remember an action and/or location. Our task design also enforces a cognitive strategy since mice are required to remember and respond to multiple randomly presented stimuli with either leftward or rightward actions, thereby circumventing postural shortcuts or other spatial forms of working memory which may form the backbone of WM in rodents [4]. Our data do not argue that mice do not have working memory when performing other tasks, just that they appear to use only RL and short term one-back strategies, not WM, when learning a contextual 2AFC task.

Our experiments also allowed us to examine mouse behavior across adolescent development. We did not observe any significant developmental changes in basic performance metrics, learning rate $\alpha$, and the softmax inverse temperature parameter $\beta$, with age. The lack of change in the RL learning rate metric and the softmax noise parameter was somewhat surprising in light of the human literature and some rodent literature, but is perhaps consistent with some existing studies of repeated discrimination learning in rodents.

Briefly, in human participants, the softmax $\beta$ parameter has been shown to (most) consistently decrease across development in humans and the $\alpha$ learning rate parameter has been shown to increase with adolescent age in some, but not all, task contexts reviewed [11, 13, 14]. Increases in RL learning rate with age have also been reported in rodents, but only for reversal learning, not initial discrimination learning [15].

In discrimination learning without reversal or before reversal, the age-performance relationship can be flat, or sometimes adolescent rodents can learn even faster than adults [7, 15]. In studies of attentional set shifting (ASST) tasks, where rodents learn a series of two choice odor-action or texture-action contingencies, adolescent and adult rats learn about new cues at a similar rate when the same modality is reinforced (intradimensional shift, IDS) but reach criterion more slowly in when the modality is changed (extradimensional shift, EDS) [15, 16]. In mice, results are more mixed with some showing no age differences in IDS and EDS performance [17], while others find adolescent mice underperform adult mice on an initial IDS or EDS but by the second IDS session adolescent learning is comparable to adults [5]. (Note that no computational modeling was reported in these set shifting studies). In the current study, the fact that we present multiple pairs of novel odorants to be associated with the same two actions and without any training on reversal makes our data most comparable to the IDS performance metric. In this light, a lack of change in fraction correct and RL learning rate in mice in the RL+WM task across development can be seen to be consistent with previous findings. Based on the needs of rodents in the wild, we speculate that learning systems must be intact just after weaning to support dispersal and the transition to independence [18].

While $\alpha$ learning rate and the softmax $\beta$ parameters did not change significantly with development in either males or females, we did find a consistently significant and positive age-related relationship in male animals in the one-back choice strategy S2 = S4: Stimulus Insensitive Win-Stay and an age and experience dependent reduction in S1 and S3 choice strategies after negative outcomes. These relationships were not significant in female mice. To test if these changes in males were dependent on developmental increases in testosterone, we examined task performance and model parameters in a cohort of mice who had undergone pre-pubertal gonadectomy or a sham surgery. We found that at young adult ages in GDX and sham groups, there were no differences in learning curves or model parameters, except for one-back strategy S3. Here the sham group was lower than seen in intact mice of the same age, suggesting a possible effect of surgery. These data suggest a rise of testosterone at puberty did

not play a causal role in any of the changes seen in one-back strategies and was not needed to support basic performance in males. Future studies might turn to other biological changes that may be sex-specific but decoupled from puberty to understand what drives changes in strategy use in males as they develop.

There were other differences between male and female mice that merit discussion. In behavior, while female mice learned the odor-action contingency within a single session, the benefit of repeat trials was lower than in males. This difference persisted in our regression models with the regression coefficient for one-back in set size = 2 not significant for females. While it is also possible that female hormones and the estrous cycle could drive changes in performance and learning strategy [19, 20], we did not measure or manipulate ovarian hormones in this study.

Here our study focused on mice, and while our findings could be extrapolated to rats there are reasons to expect rats may differ. It is possible that rats will use more working memory than mice due to inhabiting a different ecological niche and a relatively more developed prefrontal cortex [21, 22].

Another design aspect of our experiment that may be an important caveat for comparisons to others' data is the "readiness check". We started each daily session by having mice perform the task with odors A & B, only moving them on to learning novel odors when they passed a performance criterion (see Materials and methods). We chose this design to increase the chance that mice would be engaged in learning about the novel odors, given they had successfully, if briefly, indicated they were able to perform that day using task rules. However, this "readiness check" filters out participation on days with a less efficient state for learning, which may be included in other studies which do not use this method. We do not think the readiness check reduced motivation in the main task because after mice passed the readiness check they continued on to perform hundreds of additional trials per odor beyond the first 200 trials per odor that we studied here (males: 914 mean total novel odor trials, 207.18 SD; females: 987.36 mean total novel odor trials, 213.35 SD). We would argue the readiness check was unlikely to reduce WM use, because it encouraged stimulus sensitivity. Some past rodent tasks measuring WM also required the animal to work more than a hundred trials and still reported WM evidence [23].

Finally, our data add to an ongoing discussion of the multiple components contributing to laboratory task behavior in rodents. Other recent papers have argued that mice do not only rely on single strategies, but use a wide repertoire during decision making [24]. Here, we also consider how multiple strategies contribute to decision making in the context of learning, but instead of assuming that animals switch between them over the course of the experiment, we show how they can be used concomitantly throughout a learning session. Thus our approach differs from others [24–26] by looking at how multiple parallel processes contribute to behavior rather than multiple strategies supported by the same process.

## Conclusion

In conclusion, our study establishes male and female mice at different stages of development use similar strategies to learn a conditional associative 2AFC task with multiple stimuli and two alternative choices. In depth modeling of the data suggest mice do not use canonical working memory, but do use reinforcement learning and also rely on one-back strategies to solve the task.

## Materials and methods

### Ethics statement

All experimental procedures were approved by Office of Laboratory and Animal Care at the University of California, Berkeley.

### Animals

C57BL/6 mice were bred and maintained in our colony on a reverse light/dark cycle (12:12 hours) and were group housed throughout experimentation. For initial experiments, a total of 47 wild-type (WT) animals (32 male) were trained on the RL+WM task at ages ranging from P30 to P90 and were water restricted to 85–90% of baseline weight during behavioral studies. An additional 11 male mice were used for an investigation of gonadal hormone effects using pre-pubertal gonadectomy (GDX) (n = 6) or sham surgery (n = 5) and trained at ages ranging from P60 to P96.

### Gonadectomy

To test whether age differences seen in males in our initial experiments were dependent on pubertal hormones, we performed gonadectomy (GDX) or sham surgery at P25. Prior to surgery, mice were injected with 0.05 mg/kg buprenorphine and 10 mg/kg meloxicam subcutaneously and, during surgery, were anesthetized with 1–2% isoflurane at a constant flow rate. Reflexes were checked throughout the surgery and the surgery proceeded only if there were no reflexes. The incision area was shaved and scrubbed with ethanol and betadine prior to incision and ophthalmic ointment was placed over the eyes to prevent drying. A 1 cm incision was made with a scalpel in the lower abdomen across the midline to access the abdominal cavity and the testes were clamped off from the spermatic cord with locking forceps and ligated with sterile sutures. After ligation, the gonads were excised with a scalpel. The muscle and skin layers were sutured, and wound clips were placed over the incision for 7–10 days to allow the incision to heal. An additional injection of 10 mg/kg meloxicam was given both 24 and 48 h after surgery. Sham control surgeries were identical to gonadectomy except that the gonads were simply visualized and were not clamped, ligated, or excised. Mice were allowed to recover on a heating pad until ambulatory and were post-surgically monitored for 7–10 days to check for normal weight gain and signs of discomfort/distress. Mice were co-housed with 1–2 siblings who received the same surgical treatment.

### Behavioral task

Custom-built operant chambers that have an initiation port in the center and two side ports (see Fig 1A and 1B) were used for all behavioral experiments. Each port contains an infrared photodiode/phototransistor pair to accurately time port entry and exit (Island Motion, Tappan, NY). Side ports release water through valves (Neptune Research) calibrated to deliver 2 μL of water for correct choices. Slow and constant air flow ($\sim 0.5$L/min) through the center port was redirected through one of several odorant filters (Target2 GMF F2500–19) at trial initiation. Our set-up allows for the delivery of multiple odors independently, creating a seamless transition between learning programs without experimenter intervention.

For pre-training, mice were water-restricted and then trained to initiate trials via nose poke to the center port and to associate side ports with water reward for several days. Next, mice entered an early learning phase where they encountered a preliminary odor pair, A(cinnamon)—B(vanilla), which stably predicted the left and right port respectively for reward. These odors were presented for two or three days. Next, mice entered the main task where they

learned to associate novel odor pairs to a port, using deterministic reward outcomes. Mice experienced a novel odor daily if they passed a "performance readiness check" by reaching performance levels of 70% on both sides for odor pair A&B at the start of each session. In our experience, this check primes mice to use an odor-based strategy. Mice were run for at least six sessions with novel odors presented in set sizes of n = 2 or n = 4 (a minimum of three sessions per set size) for the full RL+WM task. See Supporting Information S1 Table for a list of all odor pairs used (S1 Table). If mice failed to reach this performance readiness criterion they were only exposed to odor A&B for that session. Each full task session, one per day, included 100–200 presentations of odors A&B followed by 200 presentations of each novel odor (400–800 trials depending on set size) in a pseudo-random manner, controlling for a normal delay distribution between two successive presentations of the same stimulus. All mice experienced set size = 2 prior to set size = 4, with most mice experiencing two or more sessions of set size = 2 first. Outside of daily exposure to A&B no single odor or odor pair was repeated.

## Behavioral analysis

Animal operant performance for novel odors was analyzed in MATLAB (Mathworks, Natick, MA) using custom scripts and GraphPad Prism. Three sequential sessions for set size = 2 and set size = 4 each (6 sessions total) were taken from each animal and used for learning curves and behavioral analyses as well as all subsequent regression analyses and modeling.

**Behavioral visualization and statistics.** In order to understand how mice learn to associate odor with rewarding choices at a high level, we first performed a series of behavioral analyses on the data using MATLAB (version 2018, 2021b, 2022b) and GraphPad Prism (version 9.1.1). We calculated learning curves (see Fig 2A and 2D) by taking the mean fraction correct responses out of 25 presentations of a single odor. We then took the mean of all stimuli binned in 25 trials (2 odors for set size = 2 and 4 odors for set size = 4) and plotted the mean with SEM for all mice in the group. This allowed us to compare more directly across set sizes as all odors in a session have a pseudorandom and interleaved presentation. The learning curves presented here reflect the same sessions modeled (a maximum of 3 sessions per mouse per set size). To test if there was any difference between set sizes, we validated assumptions of normality and then ran a 2-way repeated measures ANOVA in GraphPad Prism with results reported in the main text. Next, we tested whether there was a difference in current trial performance (trial $t$) when the current stimulus was repeated or non-repeated in trial $t-1$ (see Fig 2B and 2E). Starting at trial $t + 1$ for each individual session for each animal, we took the result of trial $t$ given $t - 1$ (0 for incorrect or 1 for correct) and placed the result in two new columns either for repeating or non-repeating stimuli. Then, we took the mean (fraction correct) performance for each individual animals' 3 sessions of set size = 2 and 3 sessions of set size = 4. Next we took the mean of all male or female mice and took the group mean and plotted each column for each set size with SEM. After checking assumptions for normality and variance using Anderson-Darling, D'Agostino and Pearson test, Shapiro-Wilk test, and Welsh's test, we used the appropriate comparison test (paired t-tests for within subjects and either unpaired t-tests or Mann-Whitney tests if not). Significance level was set to $p < 0.05$ for all statistical tests, and confidence intervals as well as effect sizes were reported where applicable. In Fig 2C and 2F the mean fraction correct for each session (including both set size = 2 and set size = 4) was plotted with individual animals connected by lines. To assess the impact of age on performance, separate mixed linear models were fitted for each sex using `statsmodels` package in Python. The models included age as the fixed effect predictor variable with additional fixed effects for session number and set size. Random effects (intercept and slope) were included for individual animal identity.

**Logistic regression analysis.** To better characterize trial-by-trial choice dynamics and to follow up on our behavioral results that showed that mice performed better when the stimulus repeated, we used logistic regression analysis. We developed multiple models to capture different hypotheses and interaction terms (see S1(A) Fig). In all cases, we sought to predict $P_t$(*Correct*) via a logit link function for each individual session.

In our first regression model (Regression 1), we included predictors that captured theoretical markers of incremental RL, as well as markers of strategies relying on recent information, possibly dependent on WM. The aggregated "*correct history*" regressor was meant to capture contributions of an incremental RL process and defined as the number of past correct (reinforced) choices for the current trial's odor. For WM, we examined the effect of repeating stimulus in recent past trials (1-back repeat (called *1-back* in Regression 1 for brevity) = 1 if odor on trial t, $o_t$, is the same as trial t-1, $o_{t-1}$; 0 otherwise) or two trials back (2-back repeat (*2-back*) = 1 if odor on trial t, $o_t$, is the same as trial t-2, $o_{t-2}$; 0 otherwise).

In further regression models, we considered not only the aggregated *correct history*, but more specifically the characteristics of the last trial. In particular, we were interested in whether the outcome (rewarded, no reward) and whether the odor stimulus repeated (repeat, no repeat) influenced the current choice. In addition to the first logistic regression model ("trial back") we ran the following regressions:

Regression 2, "repeat—no repeat":

- correct history, characterizing RL incremental learning.

- repeat, 1-back repeat, characterizing potential immediate performance improvement after a repeat trial. (repeat = 1, if $o_t = o_{t-1}$; 0 otherwise).

- repeat-reward, characterizing effects of previous reward in trials with repeated odor stimulus. (repeat-reward = 1 if $o_t = o_{t-1}$ and $r_{t-1} = 1$, −1 if $o_t = o_{t-1}$ and $r_{t-1} = 0$, and 0 if $o_t \neq o_{t-1}$).

- norepeat-reward, characterizing effects of previous reward in trials with a different odor stimulus. (norepeat-reward = 1 if $o_t \neq o_{t-1}$ and $r_{t-1} = 1$, −1 if $o_t \neq o_{t-1}$ and $r_{t-1} = 0$, and 0 if $o_t = o_{t-1}$).

Regression 3, "reward—no reward":

- correct history.

- reward, characterizing potential immediate performance improvement after a rewarded trial. (reward = 1, if $r_{t-1} = 1$; 0 otherwise).

- reward-repeat, characterizing a WM-like process, effects of repeated odor stimulus presentation following rewarded trials. (reward-repeat = 1 if $r_{t-1} = 1$ and $o_t = o_{t-1}$, −1 if $r_{t-1} = 1$ and $o_t \neq o_{t-1}$, and 0 if $r_{t-1} = 0$).

- noreward-repeat, characterizing a WM-like process, effects of repeated odor stimulus presentation following no reward trials. (noreward-repeat = 1 if $r_{t-1} = 0$ and $o_t = o_{t-1}$, −1 if $r_{t-1} = 0$ and $o_t \neq o_{t-1}$, and 0 if $r_{t-1} = 1$).

Note that regressions 2 and 3 are mathematically equivalent by modeling an interaction between the reward and repeat regressors; but, they distribute the interaction factor differently, providing different angles through which we can interpret the results.

Regression 4, "repeat—reward":

- correct history.

- repeat-reward, (repeat-reward = 1 if $o_t = o_{t-1}$ and $r_{t-1} = 1$; 0 otherwise).

- repeat-noreward, (repeat-noreward = 1 if $o_t = o_{t-1}$ and $r_{t-1} = 0$; 0 otherwise).

- norepeat-reward, (norepeat-reward = 1 if $o_t \neq o_{t-1}$ and $r_{t-1} = 1$; 0 otherwise).

- norepeat-noreward, (norepeat-noreward = 1 if $o_t \neq o_{t-1}$ and $r_{t-1} = 0$; 0 otherwise).

    Regression 5, "repeat—reward interaction 1":

- correct history.

- repeat, 1-back repeat. (repeat = 1 if $o_t = o_{t-1}$; 0 otherwise).

- reward, (reward = 1 if $r_{t-1} = 1$; 0 otherwise).

- reward-repeat, characterizing interaction between previous reward and odor stimulus presentation. (reward = 1 if $r_{t-1} = 1$; 0 otherwise. repeat = 1 if $o_t = o_{t-1}$; 0 otherwise).

    Regression 6, "repeat—reward interaction 2":

- correct history.

- repeat, 1-back repeat. (repeat = 1 if $o_t = o_{t-1}$; 0 otherwise).

- reward, (reward = 1 if $r_{t-1} = 1$; 0 otherwise).

- repeat-reward, characterizing effects of previous reward in trials with repeated odor stimulus. (repeat-reward = 1 if $o_t = o_{t-1}$ and $r_{t-1} = 1$, −1 if $o_t = o_{t-1}$ and $r_{t-1} = 0$, and 0 if $o_t \neq o_{t-1}$).

- norepeat-reward, characterizing effects of previous reward in trials with different odor stimulus. (norepeat-reward = 1 if $o_t \neq o_{t-1}$ and $r_{t-1} = 1$, −1 if $o_t \neq o_{t-1}$ and $r_{t-1} = 0$, and 0 if $o_t = o_{t-1}$).

Before performing regression, we performed a logarithmic transformation of the regressor *correct history* (indicating that an additional correct choice later in learning should have less impact than early in learning); then we z scored all regressors in all logistic regression models. Using AIC (Aikake Information Criterion, [27]) as well as other comparison metrics such as Bayesian Information Criterion, (BIC, [28]) as well as AICc, CAIC (data available in online repository) we determined that regression #2 and #3 performed better (had a lower score) than any other regressions (see S1(A) Fig). In regression #2 and #3, we hypothesized and aimed to resolve whether animals' choices or $P_t(Correct)$ is more driven by repeating stimuli along with specific examination of effects of previous reward in repeat or norepeat trials or by reward with specific examination of repeating odor stimulus in reward or noreward trials, respectively. However, as noted above, the two regressions are formally equivalent. Thus, we only report regression #3 in the main text (see Figs 3–5). For this model, we also tested correlations between regressors (S1(B) Fig). For a different visualization of the interaction results, we include regression #2 (S4–S6 Figs) and regression comparisons (see S1(A) Fig) in supplemental figures.

All parameters for within individuals and sessions logistic regressions were estimated using Matlab's `glmfit` and `fitglm` functions or `statsmodels` package in Python. Following normality tests on individual session regression coefficients, either parametric (one sample t) or nonparametric (Wilcoxson signed rank) tests were run in GraphPad Prism and reported in the text.

**Mixed linear model analysis.** To test the influence of development on trial-by-trial learning we determined the relationship between the age and each regression coefficient for males, females, and across both set sizes, using mixed linear models. First, we visualized and standardized the data points by taking z-scores for each session's coefficient and dropping any

outliers, namely animal sessions that were over 3 standard deviations from the mean (7 out of 96 observations for males and a similar fraction for females). Next, to coarsely test the existence of a relationship between each regression coefficient and age across sessions, we ran a simple linear regression looking at goodness of fit (determined by $R^2$ and corresponding $p$-value as shown in Figs 4 and 5) without controlling for non-independence when multiple measures were taken from a single individual. To assess the normality of each regression coefficient distribution, we employed both quantitative analysis via an Anderson-Darling normality test and qualitative examination through density and Q-Q plots. Following this evaluation, we used either a Kruskal-Wallis test or a one-way ANOVA to investigate the potential influence of animal age on the variance observed in the dependent variable (regression coefficient). After checking mixed linear model common assumptions, such as linearity and the existence of both normally distributed and independent errors, we assessed multicollinearity by calculating the Variance Inflation Factor (VIF) for the predictor variables. The VIF values were examined to ensure that there was no problematic multicollinearity in the model. The `statsmodels` package in Python 3 was utilized for this analysis. Subsequently, we fitted mixed linear models, incorporating random intercepts for each animal to control for repeated measures from each individual. Results from the mixed linear model analysis, including checks for multicollinearity, are reported in the text and/or figure legends.

## Computational modeling

**Model specifications.** To investigate the processes that support the animals' learning and decision making behavior, we used computational modeling. All models investigated included a reinforcement learning (RL) component, under the assumption that animals used reward outcomes $r = 0, 1$ to learn to estimate the value $Q(o, a)$ of making choice $a$ (left/right) for odor $o$, using a standard delta rule:

$$Q_{t+1}(o, a) = Q_t(o, a) + \alpha \times (r - Q(o, a)).$$

Values were initialized to be uninformative at $Q_0 = 0.5$. Choice was determined through a soft-max policy over estimated values: $P(a|o) \propto \exp(\beta(Q(o, a)))$, where inverse temperature $\beta$ indexed the degree of stochasticity in choosing the best option.

Beyond this baseline model, we considered whether computational models with additional mechanisms could better account for the data. Mechanisms we considered included learning rate asymmetry, forgetting, and mixture of RL with 1-back strategies. We detail these mechanisms below. Note that we did not include the WM mechanisms considered in human RL-WM models, because the very limited set size effects observed in mice indicate that human-like WM mechanisms are not at play here.

**Learning rate asymmetry**. is frequently considered in RL models and often improves fit. Here, we considered two possibilities: setting $\alpha_+$ and $\alpha_-$ as two free parameters applied to positive and negative prediction errors, respectively. Based on the results of this model where we found very low values of $\alpha_-$, we also considered a family of models with fixed $\alpha_- = 0$.

**Forgetting**. is implemented as a decay of all odor-action values towards initial values at every trial: $Q_{t+1} \leftarrow Q_t + \phi \times (Q_0 - Q_t)$, with decay parameter $0 \leq \phi \leq 1$.

**One-back strategies**. Previous work has shown that simple changes in the policy of RL models may often strongly improve fit to behavior. The simplest such strategy is a "choice kernel", or "sticky choice", whereby the mice tend to repeat the previously chosen action, independently of stimulus and outcome. This is implemented by modifying the policy such that

$$P(a|o) \propto \exp(\beta[Q(o, a) + s * I(a, a_{t-1})])$$

where $I(a, a_{t-1}) = 1$ if $a = a_{t-1}$, 0 otherwise, and $s$ is a free parameter capturing degree of tendency to repeat (or avoid) the previous trial's choice.

We additionally considered slightly more complex one-back strategies that incorporated information about the last trials' stimulus and outcome in addition to previous choice, capturing a rudimentary form of working memory. Specifically, we considered four types of trials $T_i$; $i = 1:4$, as a function of whether the previous trial was the same or a different odor and whether the previous trial was rewarded or unrewarded. Trial-type dependent strategies were expressed as a bias on the policy following:

$$P(a|o, T_i) \propto \exp\left(\beta(Q(o, a) + s_i * I(a, a_{t-1}))\right)$$

Fig 6A describes the 4 types of trials and corresponding policies. We considered various models with subsets of those policies set to 0 or to each other's values (e.g. $s_2 = s_4$) in order to simplify the model complexity. In these analyses we found that parameters $s_2$ and $s_4$ were highly correlated when fit separately, prompting us to investigate a model where they are set equal to each other, which resulted in our winning model ($s_2 = s_4$).

Our winning model (a0bs1232) thus included 5 free parameters: $\beta$, $\alpha_+$, $s_1, s_2, s_3$, as well as two fixed parameters ($\alpha_- = 0$ and $s_4 = s_2$).

**Model fitting.** We used standard maximum likelihood estimation to estimate best-fitting parameters [29]. While we considered using more state-of-the art methods such as *maximum a posteriori* or hierarchical Bayesian modeling, we found this unnecessary in this case, as the large number of trials afforded very good parameter identification (tested through a generate and recover procedure) with the simpler method, with no need for the additional assumptions that more advanced methods require (e.g. priors). We fit separately each individual and session, obtaining a set of best fit parameters for each individual and session.

**Model comparison.** We computed the BIC and also performed Bayesian model selection via the protected exceedance probability [12], using SPM12's `spm_bmf` Matlab function (Fig 6B–6D). Bayesian model selection results in a more robust model comparison than other fixed effects comparison strategies. Confusion matrices [29] confirmed that this process supported good model identifiability.

**Model validation.** To confirm that our model was able to capture mouse behavior adequately, we simulated the winning model with fit parameters (see S2 Fig), and performed similar analyses as on empirical data. For example, we were able to capture well not only group level learning curves (see S3 Fig), but also trial-by-trial dynamics at the individual and session level, as measured by the logistic regression analysis (Fig 6E and 6F).

**Mixed linear model analysis of model parameters.** To determine the relationship between age and parameter weight for each sex and set size, mixed linear models were used as explained above(Mixed linear model analysis). Both $\alpha_+$ and $\beta$ were log-transformed to $\log(\alpha_+)$ and $\log(\beta)$ prior to analysis to achieve a normal distribution and resolve issues of non-constant variance.

## Supporting information

**S1 Table. List of odor pairs used throughout the odor-based RL+WM task.** Each odor was stably paired with another odor that was similar categorically, but different enough for a mouse to discriminate between the two. With the exception of odors A & B, which was used daily for the "readiness check," no other odor pairs were repeated.
(TIF)

**S1 Fig. Logistic regression model comparison using AIC and correlation matrix for Regression #3.** We tested a series of logistic regression models that compared different aspects of

task history, trial-back identity, and interactions between various parameters (see Materials and methods for more details). (A) We found that regression #2 (described here as "repeat—no repeat") and regression #3 (described here as "reward—no reward") had the lowest AIC and, through additional analyses, determined that the two regressions were mathematically identical. Thus, we chose "reward—no reward" as the winning model and present results for this regression in the main text with regression #2 shown in the supplement. We also checked the correlation between regressors for regression #3 (B) and found that the regression intercept and the correct history parameter (here described as reg_reward_rl) as well as the main effect of reward and the interaction between reward and repeat are both the most correlated. Reward main and reward_repeat draw on the same one-back rewarded information. The negative correlation between no_reward_repeat and both correct history and the main effect of reward aligns with the two outcome choice options in our task design (reward or no reward) which oppose each other. To ensure that correlations do not affect our conclusions, we tested multicollinearity with variance impact factors (VIF) as reported in our Materials and Methods and removed any points outside of 3 standard deviations of the mean (z-scored) before we visualized or analyzed any data. (TIF)

**S2 Fig. The winning model, a0bs1232, parameters are identifiable.** We tested a series of RL-based computational models that explored which strategies mice may use to integrate and use past trial information for trial-by-trial learning. The winning model, as shown by BIC and protected exceedence probability (see Fig 6B and 6C), had 5 free parameters that included $\alpha_+$ (a), fixed $\alpha_- = 0$), a softmax $\beta$ parameter (b) and 3 one-back strategy parameters (see Fig 6A for schematic, Materials and methods for more details). In order to validate our winning model, we generated data by simulating the model with parameters fit on individual sessions. Then, we fit the simulated data in order to obtain recovered parameters. Recovered model parameters (y-axis) were highly correlated with generated parameters (x-axis; all Spearman $\rho > 0.69$, $p < 10^{-4}$), indicating that model parameters are identifiable([29]). Figure corresponds to set size = 2 males, but parameters were equally identifiable for females and set size = 4. (TIF)

**S3 Fig. The winning model, a0b1232, captures group learning for both sexes and set sizes.** In order for a model to be valid, it is important that it captures both trial-by-trial learning as well as learning that takes place across a session. Our winning model, a0b1232, includes an RL-learning component as described in Materials and methods along with 5 free parameters as described in the name. $\alpha_+$ is indicated by (a), softmax *beta* parameter by (b), the absence of any $\alpha_-$ learning rate is indicated by (0). The final model also included 3 one-back strategy parameters (S1, S2 = S4, S3, see Fig 6A for schematic) that captured how mice' responses to a current trial reflect one-back stimuli and reward experience. Here, we generated data by simulating the model with parameters fit on individual sessions. Then, we inspected the similarity between simulated data (green) and mouse data (red) and found that the model simulated with fit model parameters captures well mouse learning curve data. All data for males (A-B) and females (C-D) are binned by 25 odor presentations as shown in Fig 2A and 2D and described in Materials and methods. (TIF)

**S4 Fig. Summary of regression #2 coefficients for males and females in set size = 2 and set size = 4.** While in modeling space and in AIC comparisons (see S1 Fig) both regression #2 and regression #3 were identical, the separation of the two allowed us to better understand if the main contributor to performance was repeated trials or previously rewarded trials. All bars reflect the mean of all individual session coefficients and error bars represent SEM. For males

(A) in set size = 2, there was a significant effect of repeating trials and the interaction between repeating rewarded trials and non-repeating rewarded trials: intercept: $t = 17.74$, $df = 31$, $p < 0.0001$; correct history: $t = 14.78$, $df = 31$, $p < 0.0001$ main effect of repeat: $t = 4.7$, $df = 31$, $p < 0.0001$; repeat—reward: $t = 6.59$, $df = 31$, $p < 0.0001$; no repeat—reward: $t = 3.4$, $df = 31$, $p = 0.001$. For females (C) in set size = 2, there was only a significant effect of repeating trials when the previous trial was rewarded: intercept: $t = 10.63$, $df = 14$, $p < 0.0001$; correct history: $t = 14.61$, $df = 14$, $p < 0.0001$; main effect of repeat: $t = 1.16$, $df = 14$, $p = 0.26$; repeat—reward: $t = 1.33$, $df = 14$, $p = 0.20$; no repeat—reward: $t = 0.53$, $df = 14$, $p = 0.60$. For males in set size = 4 (B), intercept: $t = 12.34$, $df = 31$, $p < 0.0001$; correct history: $t = 13.12$, $df = 31$, $p < 0.0001$; main effect of repeat: $t = 8.15$, $df = 31$, $p < 0.0001$; repeat—reward: $t = 8.07$, $df = 31$, $p < 0.0001$; no repeat—reward: $t = 5.62$, $df = 31$, $p < 0.0001$. Finally, for females in set size = 4 (D), intercept: $t = 6.98$, $df = 13$, $p < 0.0001$; correct history: $t = 5.79$, $df = 13$, $p < 0.0001$; main effect of repeat: $t = 2.17$, $df = 13$, $p = 0.04$; repeat—reward: $t = 3.72$, $df = 13$, $p = 0.002$; no repeat—reward: $t = 4.55$, $df = 13$, $p = 0.0005$. Together these results indicate a significant influence of both correct history (RL) and combinations of one-back trials on current mouse choice. Note that for our second regression for set size = 4, the absolute number of repeat trials are fewer than set size = 2. **** $p < 0.0001$ *** $p < 0.001$, ** $p < 0.01$, * $p < 0.05$.
(TIF)

**S5 Fig. Male regression coefficients across development for regression #2 in both set sizes.** This regression tied with regression #3 for lowest AIC and sets up repeat trials as the main contributor. In order to understand the relationship between age and the predictors of current choice in trial $t$ (see S4 Fig for the logistic regression summary), we looked at whether coefficient weight (y-axis) would change over development (x-axis) for male mice. For set size = 2, (A) repeat intercept: $\beta_{age} = 0.005$, 95% CI = $[-0.002, 0.01]$, $p = 0.15$; (B) repeat correct history: $\beta_{age} = 0.002$, 95% CI = $[-0.002, 0.005]$, $p = 0.34$; (C) main effect of repeat: $\beta_{age} = 0.024$, 95% CI = $[0.016, 0.033]$, $p < 0.0001$; (D) repeat—reward: = 0.014, 95% CI = $[0.009, 0.018]$, $p < 0.0001$; (E) no repeat—reward: $\beta_{age} = -0.010$, 95% CI = $[-0.014, -0.006]$, $p < 0.0001$. For set size = 4, (F) intercept: $\beta_{age} = 0.00$, 95% CI = $[-0.006, 0.006]$, $p = 0.994$; (G) correct history: $\beta_{age} = 0.002$, 95%CI = $[-0.004, 0.008]$, $p = 0.561$; (H) main effect of repeat: $\beta_{age} = 0.008$, 95% CI = $[0.005, 0.011]$, $p < 0.0001$; and interactions between repeat stimuli and previous reward (I: repeat-reward: $\beta_{age} = 0.011$, 95% CI = $[0.004, 0.019]$, $p = 0.004$; (J) no repeat—reward: $\beta_{age} = -0.005$, 95%CI = $[-0.008, -0.002]$, $p = 0.001$. All developmental changes seen in set size = 4 were also reflected in set size = 2. Statistics and figure structures are identical to Fig 4.
(TIF)

**S6 Fig. Female regression coefficients across development for regression #2 in both set sizes.** All panels are structured similarly with the same statistical methods as Fig 5. For set size = 2, (B) repeat correct history: $\beta_{age} = -0.004$, 95% CI = $[-0.01, 0.002]$, $p = 0.20$; (C) main effect of repeat: $\beta_{age} = 0.015$, 95% CI = $[-0.008, 0.039]$, $p = 0.20$; (D) repeat—reward: $\beta_{age} = 0.002$, 95% CI = $[-0.01, 0.014]$, $p = 0.78$; (E) no repeat—reward; $\beta_{age} = -0.001$, 95% CI = $[-0.013, 0.011]$, $p = 0.86$. For set size = 4, (F) intercept: $\beta_{age} = 0.002$, 95% CI = $[-0.001, 0.013]$, $p = 0.797$; correct history(G): $\beta_{age} = 0.001$, 95% CI = $[-0.004, 0.007]$, $p = 0.605$; (H) main effect of repeat: $\beta_{age} = -0.001$, 95% CI = $[-0.009, 0.006]$, $p = 0.735$ (I) repeat—reward: $\beta_{age} = 0.003$, 95% CI = $[-0.011, 0.016]$, $p = 0.683$ (J) no repeat—reward: $\beta_{age} = 0.002$, 95% CI = $[-0.002, 0.006]$, $p = 0.337$. While it is possible that a low number of set size = 4 sessions for female mice may be responsible for no trial-by-trial learning changes over development, our findings replicate set size = 2 sessions for female mice.
(TIF)

**S7 Fig. Regression results and model validation for GDX and sham mice.** We also fit regression and computational models to the GDX and sham data. For regression #1 (A) and regression #3 (B) GDX (purple) and sham (green) sessions for individual mice for both set size = 2 and set size = 4 were averaged together and the mean of each individual was used to generate the shown group mean with SEM. Both conditions showed comparable regression weights for intercept and correct history across our two main regressions. There were a few differences between groups that reached statistical significance. In regression #1 (A), one-trial back was significantly different between age-matched GDX (n = 6) and sham (n = 5) mice ($p = 0.005$). In regression #3 (B), both reward-repeat ($p = 0.01$) and noreward-repeat ($p = 0.03$) showed differences. These differences in sham and GDX were in the same direction as differences between intact male and female mice for two of these regression coefficients (see Fig 3). However, GDX males were still statistically different from females (set size = 2 and set size = 4 pooled together for comparison to GDX). (For regression #1 (one-trial back), GDX vs. intact females: $t = 3.09$, $df = 33$, $p = 0.004$; mean ± SEM: GDX males: 0.53±0.06, intact females: 0.11 ±0.05. For regression #3 (reward-repeat), GDX vs. intact female: $t = 3.02$, $df = 33$, $p = 0.004$; mean ± SEM: GDX males: 1.07±0.06, intact females: 0.30±0.05). The last regression coefficient that was significantly different between GDX and sham, noreward-repeat, was comparable to intact female values. These data suggest GDX has subtle effects but does not phenocopy intact females. We next validated our winning model, a0bs1232 by simulating GDX/sham data and running both the simulated and mouse data through the regressions. We show simulated and mouse data for regression #3 in (I) and (J). In (C) we generated data by simulating the model with parameters fit on individual sessions. Then, we inspected the similarity between simulated data (green) and mouse data (red) and found that the model simulated with fit model parameters captures well mouse learning curve data. All data for mice and simulated data are binned by 25 odor presentations as shown in Fig 10A and described in Materials and methods. Then, for (D-H) we fit the simulated data in order to obtain recovered parameters. Recovered model parameters (y-axis) were highly correlated with generated parameters (x-axis; all Spearman $\rho > 0.69$, $p < 10^{-4}$), indicating that model parameters are identifiable [29] The main difference between sham and GDX males in modeling parameters was in S3. To test whether this was due to gonadal hormones we compared sham and GDX to age-matched intact males with both set size = 2 and set size = 4 pooled and found that sham males had significantly lower S3 values than both GDX and intact males (K)(sham vs intact: $\beta_{surgery} = -0.154$, 95% CI = $[-0.211, -0.097]$, $p < 0.0001$; GDX vs intact: $\beta_{surgery} = 0.001$, 95% CI = $[-0.061, 0.063]$, $p = 0.975$). **** $p < 0.0001$ *** $p < 0.001$, ** $p < 0.01$, * $p < 0.05$.
(TIF)

## Acknowledgments

We thank Hongli Wang and Albert Qü and other members of the Wilbrecht and Collins lab for discussion.

## Author Contributions

**Conceptualization:** Liyu Xia, Anne G. E. Collins, Linda Wilbrecht.

**Data curation:** Juliana Chase, Liyu Xia, Lung-Hao Tai, Anne G. E. Collins.

**Formal analysis:** Juliana Chase, Liyu Xia, Lung-Hao Tai, Wan Chen Lin, Anne G. E. Collins.

**Funding acquisition:** Anne G. E. Collins, Linda Wilbrecht.

**Investigation:** Juliana Chase, Liyu Xia, Lung-Hao Tai, Wan Chen Lin, Anne G. E. Collins.

**Methodology:** Liyu Xia, Lung-Hao Tai, Wan Chen Lin, Anne G. E. Collins.

**Project administration:** Anne G. E. Collins, Linda Wilbrecht.

**Resources:** Wan Chen Lin, Anne G. E. Collins, Linda Wilbrecht.

**Software:** Juliana Chase, Liyu Xia, Lung-Hao Tai, Anne G. E. Collins.

**Supervision:** Anne G. E. Collins, Linda Wilbrecht.

**Validation:** Juliana Chase, Liyu Xia, Lung-Hao Tai, Wan Chen Lin, Anne G. E. Collins.

**Visualization:** Juliana Chase, Liyu Xia, Lung-Hao Tai, Anne G. E. Collins.

**Writing – original draft:** Juliana Chase, Anne G. E. Collins, Linda Wilbrecht.

**Writing – review & editing:** Juliana Chase, Liyu Xia, Lung-Hao Tai, Wan Chen Lin, Anne G. E. Collins, Linda Wilbrecht.

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
