## [Decision Letter · Decision Letter 0]

15 Jul 2024

Dear Dr. Wilbrecht,

Thank you very much for submitting your manuscript "Adolescent and adult mice use both incremental reinforcement learning and short term memory when learning concurrent stimulus-action associations" for consideration at PLOS Computational Biology.

As with all papers reviewed by the journal, your manuscript was reviewed by members of the editorial board and by several independent reviewers. In light of the reviews (below this email), we would like to invite the resubmission of a significantly-revised version that takes into account the reviewers' comments.

As the authors will see, we received three detailed and very high-quality expert reviews. The reviewers and we agree that this paper has the potential to make an important contribution, but the many open points raised by the reviewers will need to be addressed.

We cannot make any decision about publication until we have seen the revised manuscript and your response to the reviewers' comments. Your revised manuscript is also likely to be sent to reviewers for further evaluation.

Sincerely,

Christoph Strauch

Academic Editor

PLOS Computational Biology

Daniele Marinazzo

Section Editor

PLOS Computational Biology

As the authors will see, we received three detailed and very high-quality expert reviews. The reviewers and we agree that this paper has the potential to make an important contribution, but the many open points raised by the reviewers will need to be addressed.

Reviewer's Responses to Questions

**Comments to the Authors:**

Reviewer #1: The authors were motivated to understand the extent to which working memory (WM) is utilized alongside reinforcement learning (RL) during the acquisition of stimulus-action associations. Additionally, they aimed to investigate whether WM and RL metrics undergo changes during adolescent development. Their research is primarily driven by prior human studies revealing that behavioral performance in simple instrumental tasks is more accurately described by the interplay of two processes. To determine if similar results extend to rodent models, they adapted a task previously used in humans for use with mice. They trained mice at different ages in an odor-based two-alternative forced-choice task, varying the size of the learning problem (i.e., the number of odors to associate with a left or right action) to manipulate WM effects. They conducted preliminary data analysis to inspect possible behavioral signatures of WM. Next, they performed three logistic regression models to assess the contributions of both incremental RL and WM to choice behavior. Based on the regression results, they built different computational models that extend a simple Rescorla-Wagner learning rule to assess their potential in enhancing data fit. These mechanisms include learning rate asymmetry, decay of values over time (forgetting), and the implementation of strategies based on previous information (one-back strategies), including the tendency to repeat the previous action or incorporating information about the trial history. Finally, leveraging the longitudinal nature of their design, the authors examined the effect of age on task and model metrics. Overall, I believe the work addresses a significant topic by validating human paradigms in the murine model, which holds paramount importance due to its genetic manipulability. Nevertheless, this current version of the manuscript has several major issues that the authors need to address.

Reviewer #2: Summary

The authors contribute to computational modeling of behavior by examining the interplay between reinforcement learning (RL) and working memory (WM) in mice performing a 2AFC odor discrimination task using interleaved odor set sizes of either 2 or 4 novel odors. They also examine how utilization of these strategies changes with age, gender, and odor set size. Their goal was to establish a comparative baseline between humans and mice when measuring RL+WM, and whether mice show similar strategy changes between set sizes and throughout development. Using two modeling approaches, they find that mice employ both RL and a simple form of WM captured as 1-trial-back perseveration dependent on previous odor and reward, without finding significant differences between odor set sizes. They report age-related changes in the utilization of WM, but these changes are inconsistent across male and female mice. They don’t find age-related changes in RL in either gender. These results contrast with findings in humans, highlighting differences between the two species.

Overall, I find the work to be an important contribution towards understanding species differences in the deployment of RL vs WM strategies, especially when mouse models are frequently used as a proxy to study similar mechanisms in humans. However, the modeling of RL alongside other strategies is not itself a novel contribution (e.g. Miller et al., 2017, Nature Neuroscience; Akam et al., 2021, Neuron; Miller et al., 2022, eLife; and Venditto et al., 2024, eLife, all model strategy mixtures combining RL with perseverative agents similar to the RL+WM model used here); although the specific setting in which the model is applied here appears novel. Additionally, their specific parameterization raises some concern over how its interpreted (major issue below). The manuscript would also benefit from additional clarification and justification (minor issues below).

Major issues

• RL+WM model parameterization: Although the equation before line 190 is missing a closing parenthesis, given the equation before 188, it is fair to assume that the closing parenthesis should be at the end. In both equations, then, the softmax beta acts on both the RL Q(o,a) as well as the choice kernel I(a,at-1). In this parameterization, beta acts as a decision noise parameter on all choice-related information and should not be interpreted as a weight on the RL component. This raises concern over the interpretation of data in figures 7B,G, 8B,G, and 9B,G, which does not fully represent changes in RL strategy across age or sessions. Furthermore, in all these figures I find the log-scale to be slightly misleading towards the true variation in beta weights, which seems strikingly large. Many of the weights seem to hit a ceiling at log(3)=20, which also appears to be an issue in the parameter recovery data shown in figure S1, suggesting that these fitted values are unreliable. It is possible that parameter interactions between beta and the WM weights are contributing to this fitted behavior. Another contributor may be the lack of modeling a bias (Wilson & Collins 2019).

Minor issues

• Materials and methods: Behavioral task: there are some details missing that inhibit replicability, as well as some things that need further clarification

- What is the total number of odors used, and what odors were they?

- Were any odors reused outside the preliminary odor pair, or was it a novel set in every new session?

- How were odor set sizes interleaved?

- On line 83-84, does “six sessions total per mouse” include the two to three training sessions, or does this mean three sessions of each set size, leading to six sessions in addition to early learning?

- It might be helpful in Figure 1 to give an example of odor sequences presented within a single session to clearly lay out the A/B odor phase and transition to novel odor set

• Results: Behavioral task and results: While a second description of the task is useful in this section, including details of pre-training feels redundant as it doesn’t have as much bearing on the results.

• Logistic regression analysis and models: I feel the selection of predictors for regressions 2 and 3 could be better motivated, and the benefit of splitting it out this way is unclear. Neither regression allow for direct comparison between main effects of cues and rewards (as would p ~ 1 + repeat + reward + repeat-reward), nor do they allow direct comparison of possible interactions (as would p ~ 1 + repeat-reward + norepeat-reward + repeat-noreward). They’re also technically the same model, where one can be written as a linear combination of the other (e.g. reward = repeat-reward + norepeat-reward; noreward-repeat = repeat – repeat-reward). This also means the coefficients on correct history should be identical across the two models, which appears to be the case in Figure 3, Figure S3 (despite different scales), Figure 4B,G, Figure S4B,G, Figure 5B,G, and Figure S5B,G. These panels should not be treated as separate results.

- In methods lines 116-117, the justification is “to capture different hypotheses”, but doesn’t explain what those hypotheses are. Also, both regressions 2 and 3 should be outlined here, not just 3.

- In results lines 297-298, the justification is to “triangulate the potential interaction between repeated cues and reward”, but neither model allow for direct comparison between main effects of cues and rewards nor all interactions.

- The Figure 3 caption states that regression 2 captures “the influence of stimulus repetition” and 3 captures the “influence of past trial’s reward”, but it’s unclear how these models separate the two when both models include past influence of both stimulus and reward, just with a different parameterization.

• Materials and methods: One-back strategies: Although it was clarified in the results, the mapping between trial type (repeat or non-repeat, rewarded or unrewarded) to trial number (1-4) is laid out in the methods. Therefore, taking this section by itself, it’s unclear why setting s2=s4 is a valid pairing.

• Results: Computational Modeling: The name associated with S1, “Inappropriate Lose-Shift”, appears confusing because, by construction, all one-back strategies should reinforce making the same choice. Shouldn’t S1 bias the policy towards repeating choices after a stimulus change and previously unrewarded choice? Or has the weight on S1 been flipped prior to plotting to correspond with choice switching?

• Figures: For many of the bar plots, it is unclear from the figure legend what is being plotted (mean or median), what the error bars are computing (e.g. SEM, 95% CI), and what each are being computed over (rats or sessions). Some figure specific comments:

- Figure 3,S3: it would be useful to have asterisks to visually identify significant bars

- Figure 4,5,7,8: Putting asterisks next to significant p-values would help identify significant regressions

- Figure 6C,D: Make explicit what “exp_r” and “pxp” is in the figure legend

- Figure 7: What is the red box around panels D and I?

- Figure S1: What are the correlation coefficients for each panel? It would be useful to have this information overlaid on each plot

• Line 117: “logistic link function” should be “logit link function”

• LaTeX formatting: Most of the opening quotation marks are backwards. There are also redundant parentheses around in-text citations.

Reviewer #3: Chase and colleagues present a study investigating the use and development of different learning strategies from adolescence to adulthood in mice. Adapting a task developed in humans that can be solved, to different extents, by reinforcement learning and working-memory based mechanisms mice learned odor-based conditional associations in small or large set sizes. The authors analysed the data at the gross performance level, as well as using regression analyses and computational models. The results suggest, contrary to humans, that mice rely more on RL strategies, with little evidence of a clear WM influence on task performance. While there was evidence that male mice performed better when there were fewer items to learn this was related to an increased reliance on repeating a choice after a reward (win-stay), which was both more likely to happen in the smaller set size, and increased across age.

As the paper stands I believe it could make a valuable contribution to the literature and the justification behind the work in providing a greater knowledge of the value of mice models to understanding some aspects human cognition and cognitive dysfunction is hugely important. I believe the study design was a pragmatic adaptation from the human work for use in mice, and the analyses are comprehensive and carefully applied.

However, my review focuses on the interpretation of the results in the context of the challenges in translating tasks between humans and non-human animals. While a direct answer to these points is not possible without conducting additional experiments (which I don’t necessarily suggest), the results of such future work could reduce our confidence in the current paper’s conclusions by challenging the results.

For example, the inclusion of the readiness check requires the animal to complete up to 200 trials before the main experiment. The authors, already point to this as a difference between the present study and other past work that suggest mice can use WM-based strategies. However, the authors could do more to explore the potential mechanisms that this difference may affect and the impact of that specifically on their current result. One potential difference the readiness check introduces is fatigue which likely reduces WM capacity. Another would be motivation, as by the time the mice reach the main experiment they will have relatively more of their daily water allowance compared to mice in other experiments. While these internal states may seem minor experimental differences they may shift the balance of influence of different learning strategies, making it appear that mice are less able in this context to use WM-based mechanisms. I think this weakness could be more clearly considered in the discussion.

Similarly, the differences between male and female mice are intriguing and could be discussed in greater detail. The current interpretation that the increase in win stay strategy was related to the greater energy cost in moving males’ heavier bodies feels very speculative. If the authors feel strongly about this point, I would appreciate more evidence to link these two things together.

On a related point, a potential unexplored methodological difference that may explain the lack of developmental effects in the females is that puberty is not considered in this study. There is evidence that female mice, like humans, go through puberty at an earlier age than males. It is therefore possible that the key developmental window was missed in females in this study. While this would not explain the overall male/female difference a developmental shift still may occur in both species. I suggest this potential issue could be noted in the discussion.

A final, very minor point is that I believe that many of the figure legends contain too much detail and may be repetitive. Its unusual to include a lot of stats results in main text figures, particularly if they overlap with what’s included in the manuscript body. Furthermore, the structure of the figures repeats between male and female so the repetitive summaries could be shorted to “figure format is identical to the males”, for example.

**Have the authors made all data and (if applicable) computational code underlying the findings in their manuscript fully available?**

Reviewer #1: Yes

Reviewer #2: None

Reviewer #3: None

PLOS authors have the option to publish the peer review history of their article (what does this mean?). If published, this will include your full peer review and any attached files.

Reviewer #1: No

Reviewer #2: No

Reviewer #3: **Yes: **MaryAnn Philomena Noonan
---

## [Decision Letter · Decision Letter 1]

22 Nov 2024

Dear Dr. Wilbrecht,

We are pleased to inform you that your manuscript 'Adolescent and adult mice use both incremental reinforcement learning and short term memory when learning concurrent stimulus-action associations' has been provisionally accepted for publication in PLOS Computational Biology.

We also suggest that you take into account the comments of Reviewer 2.

Best regards,

Lyle J. Graham

Section Editor

PLOS Computational Biology

Daniele Marinazzo

Section Editor

PLOS Computational Biology

Feilim Mac Gabhann

Editor-in-Chief

PLOS Computational Biology

Jason Papin

Editor-in-Chief

PLOS Computational Biology

Reviewer's Responses to Questions

**Comments to the Authors:**

Reviewer #1: The authors have addressed all the comments thoroughly, and the revisions meet the standards for publication.

Reviewer #2: I believe the authors' revisions sufficiently address mine and the other reviewers' comments, while enhancing the overall clarity and impact of the paper. The additional study testing the impact of hormones on the age-related effects seen in males was the cherry on top. Despite a negative result, I believe it to be a valuable addition in understanding the present results and for steering future studies.

I have a couple very minor points that I believe may be useful but, by themselves, don't merit another round of review:

- On line 176, the sentence beginning with "In regression #3" is incomplete and has mismatched parentheses.

- A new result highlighted in the main text (line 320-321) and the Figure 10 caption (last sentence) points out a significant difference between the sham mice and the intact mice, but this difference isn't visualized in Figure 10 or in any supplemental figure. It would be a nice addition to have this result visualized (either in Figure 10 or in an additional supplement)

- Supplemental figure 7 should include a color legend in panels A and B (although it is mentioned in the figure caption)

Reviewer #3: The authors have carefully considered my comments and have made substantial changes to the manuscript in light of my and the other reviewers. In particular their use of the inclusion of the additional study to investigate whether male sex hormones play a role in their effects provide a relatively conclusive answer that puberty does not impact male mice behavioural strategies.

I am happy to support the manuscript to publication.

**Have the authors made all data and (if applicable) computational code underlying the findings in their manuscript fully available?**

Reviewer #1: Yes

Reviewer #2: None

Reviewer #3: None

PLOS authors have the option to publish the peer review history of their article (what does this mean?). If published, this will include your full peer review and any attached files.

Reviewer #1: No

Reviewer #2: No

Reviewer #3: **Yes: **MaryAnn Noonan

---

## [Editor Report · Acceptance letter]

15 Dec 2024

PCOMPBIOL-D-24-00791R1 

Adolescent and adult mice use both incremental reinforcement learning and short term memory when learning concurrent stimulus-action associations

Dear Dr Wilbrecht,

I am pleased to inform you that your manuscript has been formally accepted for publication in PLOS Computational Biology. Your manuscript is now with our production department and you will be notified of the publication date in due course.

With kind regards,

Dorothy Lannert
